# EUREC⁴A observations from the SAFIRE ATR42 aircraft

Sandrine Bony[1], Marie Lothon[2], Julien Delanoë[3], Pierre Coutris[4], Jean-Claude Etienne[5],
Franziska Aemisegger[6], Anna Lea Albright[1], Thierry André[8], Hubert Bellec[8], Alexandre Baron[7],
Jean-François Bourdinot[8], Pierre-Etienne Brilouet[2,5], Aurélien Bourdon[8], Jean-Christophe Canonici[8],
Christophe Caudoux[3], Patrick Chazette[7], Michel Cluzeau[8], Céline Cornet[9], Jean-Philippe Desbios[8],
Dominique Duchanoy[8], Cyrille Flamant[10], Benjamin Fildier[11], Christophe Gourbeyre[4],
Laurent Guiraud[8], Tetyana Jiang[8], Claude Lainard[8], Christophe Le Gac[3], Christian Lendroit[8],
Julien Lernould[8], Thierry Perrin[8], Frédéric Pouvesle[8], Pascal Richard[5], Nicolas Rochetin[11],
Kevin Salaün[8], Alfons Schwarzenboeck[4], Guillaume Seurat[8], Bjorn Stevens[12], Julien Totems[7],
Ludovic Touzé-Peiffer[1], Gilles Vergez[8], Jessica Vial[1], Leonie Villiger[6], and Raphaela Vogel[1]

[1]LMD/IPSL, Sorbonne Université, CNRS, Paris, France
[2]Laboratoire d'Aérologie, University of Toulouse, CNRS, Toulouse, France
[3]LATMOS/IPSL, Université Paris-Saclay, UVSQ, Guyancourt, France
[4]LAMP, Université Clermont Auvergne, CNRS, Clermont-Ferrand, France
[5]CNRM, University of Toulouse, Météo-France, CNRS, Toulouse, France
[6]Institute for Atmospheric and Climate Science, ETH Zurich, Zurich, Switzerland
[7]LSCE/IPSL, CNRS-CEA-UVSQ, Université Paris-Saclay, Gif-sur-Yvette, France
[8]SAFIRE, Météo-France, CNRS, CNES, Cugnaux, France
[9]LOA, Université de Lille, CNRS, Lille, France
[10]LATMOS/IPSL, Sorbonne Université, CNRS, Paris, France
[11]LMD/IPSL, École Normale Supérieure, CNRS, Paris, France
[12]Max Planck Institute for Meteorology, Hamburg, Germany

**Correspondence:** sandrine.bony@lmd.ipsl.fr

**Abstract.** As part of the EUREC⁴A (*Elucidating the role of cloud-circulation coupling in climate*) field campaign, which took place in January and February 2020 over the western tropical Atlantic near Barbados, the French SAFIRE ATR42 research aircraft conducted 19 flights in the lower troposphere. Each flight followed a common flight pattern that sampled the atmosphere around the cloud-base level, at different heights of the subcloud layer, near the sea surface and in the lower free troposphere.

The aircraft's payload included a backscatter lidar and a Doppler cloud radar that were both horizontally oriented, a Doppler cloud radar looking upward, microphysical probes, a cavity ring-down spectrometer for water isotopes, a multiwavelength radiometer, a visible camera and multiple meteorological sensors, including fast rate sensors for turbulence measurements. With this instrumentation, the ATR characterized the macrophysical and microphysical properties of trade-wind clouds together with their thermodynamical, turbulent and radiative environment. This paper presents the airborne operations, the flight

segmentation, the instrumentation, the data processing and the EUREC⁴A datasets produced from the ATR measurements. It shows that the ATR measurements of humidity, wind and cloud-base cloud fraction measured with different techniques and samplings are internally consistent, that meteorological measurements are consistent with estimates from dropsondes launched from an overflying aircraft (HALO), and that water isotopic measurements are well correlated with data from the Barbados Cloud Observatory. This consistency demonstrates the robustness of the ATR measurements of humidity, wind, cloud-base



cloud fraction and water isotopic composition during EUREC[4]A. It also confirms that through their repeated flight patterns, the ATR and HALO measurements provided a statistically consistent sampling of trade-wind clouds and of their environment. The ATR datasets are freely available at the locations specified in Table 11.

## 1 Introduction

The interaction of trade-wind clouds with their environment is at the center of fundamental questions such as the role of clouds in climate sensitivity. The EUREC[4]A field campaign, which took place in Jan-Feb 2020 near Barbados, has been designed specifically to address this issue (Bony et al., 2017). During one month, four research aircraft, four research vessels, ground-based observations and a myriad of autonomous observing systems characterized clouds and the environment surrounding them over a large range of space scales (Stevens et al., 2021). To elucidate the couplings between clouds and circulation, the

nucleus of the experimental strategy was based on the coordinated and repeated flight plans of two core platforms : the *High Altitude and Long-range Research Aircraft* (HALO) operated by the German Aerospace Center (Konow et al., 2021), and the ATR-42 (hereafter referred to as ATR) operated by the French *Service des Avions Français Instrumentés pour la Recherche en Environnment* (SAFIRE). These airborne operations were augmented with other platforms operating within the same area, including the Twin Otter operated by the British Antarctic Survey (Blyth et al, in prep), the P-3 aircraft operated by NOAA

(Pincus et al., 2021), a Barbadian aircraft operated by the Regional Security System (RSS), the BOREAL and Skywalkers UAVs operated by Météo-France, and the CU-RAAVEN UAV operated by the University of Colorado (de Boer et al., 2021). In addition, ground and ship-based observations from the Barbados Cloud Observatory (Stevens et al., 2016) and a research vessel (R/V Meteor) were continuously documenting the atmospheric state on the western and eastern sides of the ATR operations area, respectively.

While HALO was flying at an altitude near 10 km to observe the cloud field from above and to document the environment of clouds with dropsondes, the ATR was flying in the lower troposphere to characterize clouds and their environment through *in-situ* and remote sensing measurements. To help understand the physical processes that control the climate change cloud feedbacks and the mesoscale organization of shallow convection, the primary mission of the ATR was to measure the cloud fraction near cloud base and the dynamical and thermodynamical environment of clouds from the turbulent scale to the meso

scale (Bony et al., 2017).

Due to the nature of the trade-wind regimes, fullfiling this mission constitutes an experimental challenge. First of all, the cloud field in these regimes is composed of very small and thin broken clouds, with an expected cloud fraction at cloud base of only a few percent. Accurate measurements of the cloud-base cloud fraction therefore require both a good sensitivity of the instruments to the presence of clouds, and an adequate sampling of the cloud field. Secondly, the humidity field is associated

with extremely large and steep vertical gradients, ranging from 80% near the surface, to 100 % within clouds, to less than 5 %





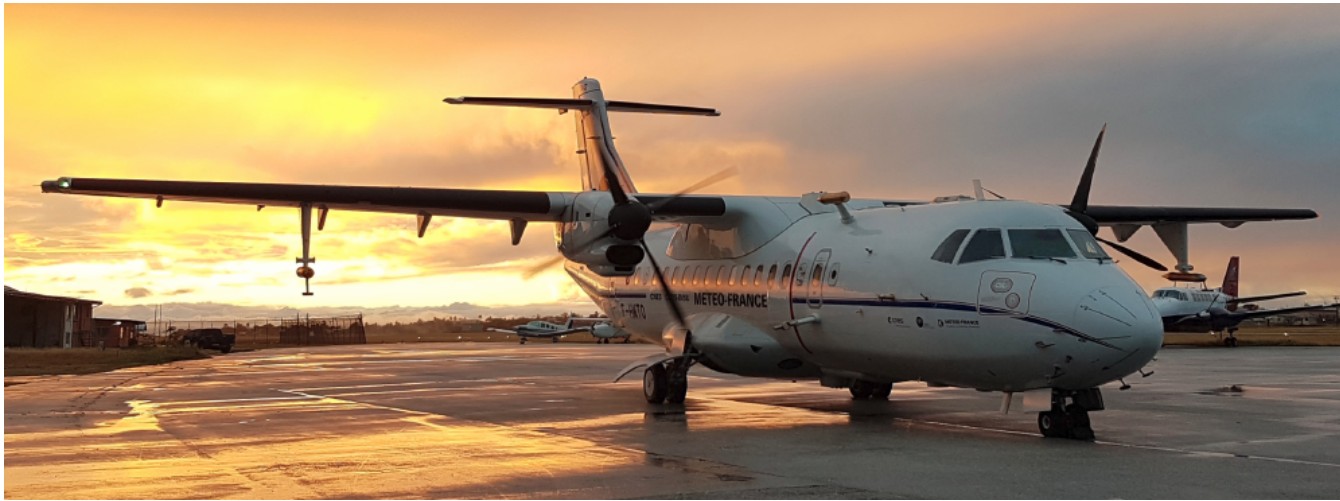

**Figure 1.** The ATR coming back from its successful EMI flight in Barbados on Jan 23 2020.

above the trade inversion (Stephan et al., 2021). These gradients favour phase changes and the deposition of cloud droplets on airborne sensors, which can affect the response time and accuracy of the measurements.

These challenges were met by fitting the aircraft with a wealth of instrumentation which, in some cases, was used in an airborne configuration for the very first time. The instrumentation was also chosen to promote redundancy or complementarity of sensors and measurement techniques. This redundancy was not only important for the post-processing and calibration of the data, it was also essential to assess the robustness of the ATR measurements of cloud fraction, humidity and winds.

The goal of this paper is to provide an overview of the operations and measurements of the ATR during EUREC$^4$A. Section 2 presents the aircraft, the operations, the flight patterns and their segmentation, and the weather conditions during the flights. Section 3 presents the ATR instrumentation, ranging from the core instrumentation of the aircraft to the instruments that were specifically devised for EUREC$^4$A, and provides a brief description of the data post-processing and of the associated datasets. The focus is put on the datasets which have not been subject to specific data papers. Section 4 assesses the internal consistency of ATR measurements regarding the cloud-base cloud fraction, humidity and wind, and their consistency with observations from other platforms. Links to the data are provided in Section 5 and a brief summary and conclusions are presented in Section 6.



## 2 Flights and operations

### 2.1 A challenging start

The SAFIRE ATR42 (F-HMTO) is a turbo-propeller aircraft flying in the lower troposphere (its ceiling is at about 7.5 km) which has been modified in many ways to fit scientific research purposes. The preparation of the ATR for the EUREC[4]A campaign was associated with significant challenges.

First of all, the ATR homebase is in Toulouse (in the south of France) and to join the Carribean during boreal winter, the aircraft had to follow the historical route of the *Aerospostale* through Tenerife (Canary islands), Prahia (Cape Verde) and Fortaleza (Brazil). As the crossing of the Atlantic ocean required an exceptionally long flight (8 hours) compared to the maximal autonomy of the aircraft (8.5 hours), the ATR had to be kept as light as possible during the transit. For this purpose, most of the EUREC[4]A instruments and aircraft equipment had to be unmounted from the ATR in Toulouse and shipped to Barbados well in advance of the transit. The final segment from Fortaleza and Barbados was also at the limit of the aircraft autonomy. Unfavourable wind conditions imposed a refuel in Cayenne (French Guyana), but the ATR finally landed in Barbados five days after its departure from Toulouse. It was the most remote campaign ever accomplished by this aircraft.

Second, extraordinary circumstances independent of SAFIRE and EUREC[4]A considerably delayed the maintenance and the upgrade of the aircraft avionics during the last months before the campaign. As a result, and for the first time in SAFIRE history, the full integration of the campaign's scientific payload into the aircraft could not take place in Toulouse as planned but had to be accomplished on site. Most of the aircraft equipment and scientific instruments were mounted on the aircraft after the ATR landed in Barbados on Jan 19, and the whole EUREC[4]A payload flew for the first time in the ATR during the Electromagnetic interference (EMI) flight which took place in Barbados on Jan 23. Although the EMI test was successful, this first flight with the whole EUREC[4]A instrumentation revealed a number of problems that had to be fixed. Therefore, the ATR did not participate in the first coordinated flight of the EUREC[4]A campaign on Jan 24 but planned another test flight on Jan 25 (RF02), including special maneuvers for calibration purposes, and started coordinated missions with the other aircraft on Jan 26. On Jan 26 unfortunately, the Inertial Navigational System (INS) of the scientific instruments showed malfunctioning. A solution was found, requiring however that for the rest of the campaign, the acquisition rate of navigation data be recorded at 50 Hz instead of 100 Hz.

Despite these challenges to prepare the aircraft for the campaign, the ATR conducted 19 research flights on 11 operation days from Jan 25 to Feb 13 2020 (totalling approximately 82 flight hours, Table 1), and successfully fullfilled the scientific mission that it aimed to accomplish.

### 2.2 Flight patterns

The ATR generally performed two flights per day in coordination with the other aircraft. Each research flight was typically 4.5 or 5 hour long, including a transit time from the airport to the EUREC[4]A circle of about 20 min in each direction. The refuel

| ATR flight | Date MM-DD | Take-off UTC | Landing UTC | R-pattern $R_b$/$R_t$: cloud base/top | L-pattern subcloud-layer | S-pattern near-surface | Ferry legs above clouds | Comment |
|---|---|---|---|---|---|---|---|---|
| RF02 | 01-25 | 13:43 | 17:42 | 623 m ($2R_b$) | 537, 345 m | | 4835,4505,2260,2905 m | test, no HALO |
| RF03 | 01-26 | 11:59 | 16:04 | 780 m ($2R_b$) | 579, 404 m | 62 m | 2575, 4510 m | INS failure |
| RF04 | 01-26 | 16:57 | 21:26 | 817 m ($3R_b$) | 613, 398 m | 64 m | 2570, 1600, 965 m | |
| RF05 | 01-28 | 20:36 | 00:50 | 669 m ($3R_b$) | 495, 328 m | | 3215, 2555, 1925 m | night landing |
| RF06 | 01-30 | 11:11 | 15:31 | 711 m ($3R_b$) | 526, 301 m | 62 m | 2575, 1625, 640 m | HALO on R[1] |
| RF07 | 01-31 | 14:59 | 18:48 | 630 m ($3R_b$) | 383, 203 m | | 2605, 3235 m | |
| RF08 | 01-31 | 19:49 | 00:01 | 613 m ($3R_b$) | 464, 306 m | | 2585, 3240 m | night landing |
| RF09 | 02-02 | 11:34 | 15:37 | 775 m ($2R_b$) | 621, 314 m | | 2575, 3230, 1105 m | |
| RF10 | 02-02 | 16:44 | 21:03 | 781 m ($3R_b$) | 608, 307 m | 61 m | 3220, 3220, 2580 m | |
| RF11 | 02-05 | 08:45 | 12:59 | 1900,746 m ( $R_t$, $2R_b$) | 552, 276 m | 64 m | 3215, 3235 m | night takeoff |
| RF12 | 02-05 | 13:48 | 18:04 | 790 m ($3R_b$) | 513, 235 m | 66 m | 2265, 3225 m | |
| RF13 | 02-07 | 11:30 | 15:51 | 2128,1051 m ($R_t$, $2R_b$) | 615, 316 m | 65 m | 2585, 3230 m | |
| RF14 | 02-07 | 17:20 | 21:42 | 855 m ($3R_b$) | 659, 324 m | 61 m | 2570, 3210 m | |
| RF15 | 02-09 | 08:37 | 13:08 | 822 m ($3R_b$) | 621, 304 m | 63 m | 3210, 4510 m | night takeoff |
| RF16 | 02-09 | 14:03 | 18:23 | 792 m ($4R_b$) | | 68 m | 2600, 4495 m | RSS[2] |
| RF17 | 02-11 | 05:55 | 10:21 | 1863, 717 m ( $R_t$, $2R_b$) | 583, 273 m | | 4495, 2570 m | P3[3], night flight |
| RF18 | 02-11 | 11:30 | 15:51 | 774 m ($3R_b$) | 551, 279 m | 66 m | 4035, 2420 m | |
| RF19 | 02-13 | 07:35 | 11:52 | 1985, 801 m ($R_t$, $2R_b$) | 600, 303 m | 69 m | 2250 m | night takeoff |
| RF20 | 02-13 | 13:14 | 17:37 | 863 m ($2R_b$) | 604,297,154 m | 69 m | | INS failure |

**Table 1.** List of ATR flights with a brief description of the main flight patterns: the mean approximate height (and number) of rectangles flown around cloud-base ($R_b$) or cloud-top ($R_t$), the height of the L-patterns flown near the top and the middle of the subcloud-layer, the height of the near-surface leg (S-pattern) and of the Ferry legs flown above clouds. [1] On Jan 30 2020, from 11:42 to 12:32 UTC, HALO flew two race-track patterns above the ATR rectangle. [2] On Feb 9 2020, from 14:32 to 17:00 UTC, the ATR flew within the field of view of the RSS aircraft. [3] On Feb 11 2020, from 4:17 to 7:25 UTC, the P3 flew two circular patterns within the EUREC[4]A circle at an altitude of about 7.5 km and dropped 12 sondes along its first circle (from 4:17 to 5:55 UTC) just before the ATR take-off.

in Barbados between two flights was about one hour long, so that within 90 min, the ATR was back in the measurement zone for a second mission (Table 1). While the ATR was flying in the lower troposphere, HALO was observing the cloud field from aloft and was droping sondes along three consecutive circles of about 200 km diameter (Konow et al., 2021).

The ATR's mission was primarily focused on characterizing the cloud-base cloudiness, subcloud-layer properties and their signals of spatial organisation at the turbulent scale and at the mesoscale. For this purpose, each flight was composed of a basic set of patterns near cloud-base and within the subcloud-layer that was repeated independent of meteorological conditions. This repetition was motivated by the wish to sample the diversity of boundary layer conditions without any bias, and to compare

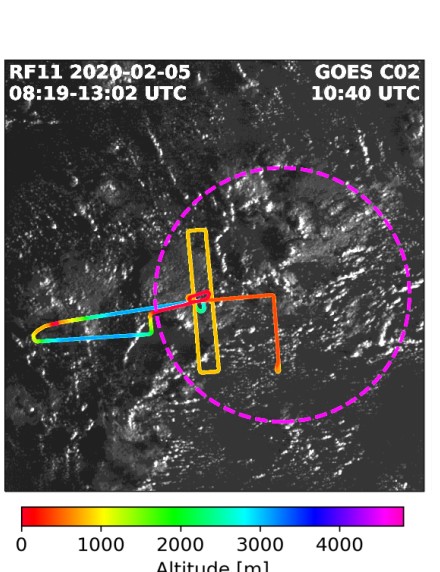

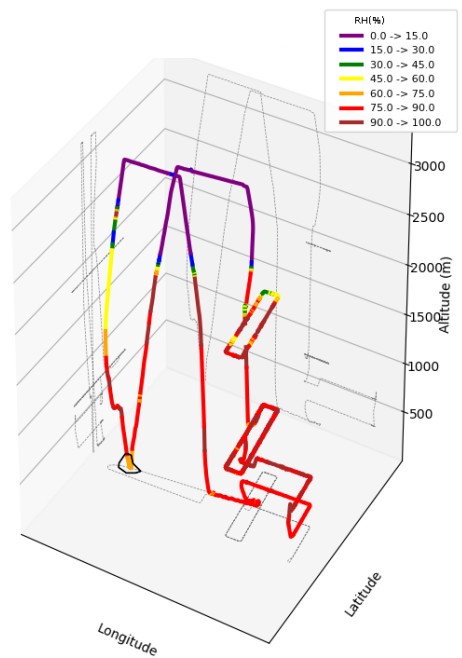

**Figure 2.** (left) Longitude-latitude trajectories of the ATR coloured by the flight altitude (for repetitive flight patterns such as the rectangles, only the last repetition is visible due to overlap). The dashed circle represents the EUREC[4]A circle. The ATR track is shown for RF11 (Feb 05) on top of a satellite snapshot of the domain (57-60$^o$W, 11.8-14.8$^o$N) derived from the visible channel of GOES-16 at about mid-flight time (the tracks of all other flights are shown on Fig. B2). (right) 3D representation of the ATR trajectory during the same flight (RF11), colored by the relative humidity measured at the flight level.

the flights with each other. Then, depending on flight and weather conditions, a few additional patterns were flown near cloud
top, at cloud-base and/or near the sea surface. Owing to the sharp vertical humidity gradients of the atmosphere and the need
to minimize the instruments' memory effects, and due to the abundant presence of sea salt near the ocean surface which can
dirty the instruments'optics, the patterns were preferentially flown from top to bottom.

Shortly after takeoff, the ATR ferried towards the EUREC[4]A circle generally at an altitude of 2.5, 3.5 or 4.5 km, so above or
around the trade inversion level (Fig. 2). Once arrived over the measurements area, it started to fly large rectangles (or race-track
patterns, also referred to as 'R-patterns') of about 120 km × 20 km, perpendicular to the mean Easterly wind. The width of the
rectangle was chosen so as to best sample the cloud field within the rectangle area using horizontal lidar-radar measurements
(section 3.5.3). At least two rectangles were flown around cloud base (around 750 m), at an altitude determined with the help
of the ground-based support (section 2.3). When an extensive stratiform cloud layer was present near the trade-inversion level
(as during RF11, RF13, RF17 and RF19), the ATR could fly an additional rectangle around cloud top (near 2 km). Otherwise
it flew an additional rectangle at cloud-base to increase the cloud-base sampling. The flight trajectories and patterns associated
with each flight are shown in Figs. B1 and B2.





Then, to characterize the turbulent and mesoscale organization of the subcloud-layer, the ATR flew two L-shape patterns within the subcloud-layer, one near the top of the subcloud-layer (generally around 600 m) and the other near the middle of the subcloud-layer (around 300 m). As the organization of the boundary layer can be anisotropic and dependent on the wind direction, each L-pattern was composed of two straight legs perpendicular to each other (each leg being about 60 km long): one along-wind and one cross-wind. Finally, in daylight conditions a near-surface leg of about 40 km was performed at an altitude of about 60 m before returning to the Grantley Adams International Airport (BGI) in Barbados through another Ferry leg in the free troposphere.

A few flights were associated with particular features:

- During RF06 (Jan 30), from 11:42 to 12:32 UTC, HALO flew (twice as fast as the ATR) two race-track patterns above the ATR rectangle at an altitude of about 10 km; two dropsondes were dropped at the extremities of the HALO race-track. This coordinated flight will help compare the cloud detection and characterization performed with the HALO and ATR measurements.

- During RF16 (Feb 9th), the ATR flew within the field of view of the Regional Security System (RSS) aircraft, which was flying parallel to the ATR at about the same altitude. On this occasion, the ATR flew 4 rectangles around cloud base. The coordination between the two aircraft will help compare the cloud detection performed with the ATR instruments with the high-resolution pictures taken by the visible camera of the RSS aircraft.

- During RF17 (Feb 13th), the ATR flew during night time. This flight was coordinated with the P-3 aircraft (Pincus et al., 2021), that dropped sondes (from an altitude of about 7.5 km) along the EUREC$^4$A circle right before the ATR take-off.

## 2.3 Ground support

The main role of the ATR during EUREC$^4$A was to measure the cloud fraction and the thermodynamical, dynamical and microphysical properties of the atmosphere at the interface between the subcloud layer and the cloud layer (Bony et al., 2017; Stevens et al., 2021). A ground crew estimating cloud base height using real-time observations from several observing platforms near and within the targeted flight area provided tactical support for each flight mission. It advised the flight planning about the cloud-base level and about the relevance of flying at the top of the cloud layer when an extensive layer of stratiform cloudiness was present near the trade inversion.

As illustrated by Fig. 3, the targeted cloud-base level was not the lifting condensation level (LCL) but the height of the maximum near-base cloud fraction ($z_{CFmax}$). This level corresponds to the level where most clouds in the sampling area have reached their base level, and it is most adequately defined by the height at which a cloud radar reports a maximum cloudiness near cloud base. The cloud-base height distributions from the ceilometer and estimates of the mixed layer top, sub-cloud layer top ($h$), and lifting condensation level (LCL) from soundings and surface weather data provided further guidance for choosing the correct cloud base level.

The evening before the flight, and again 2 hours before take-off, a pre-flight estimation of the flight levels was performed based on near real-time cloud radar, ceilometer, radiosonde and surface weather data from the Barbados Cloud Observatory



Earth System
Science
Data

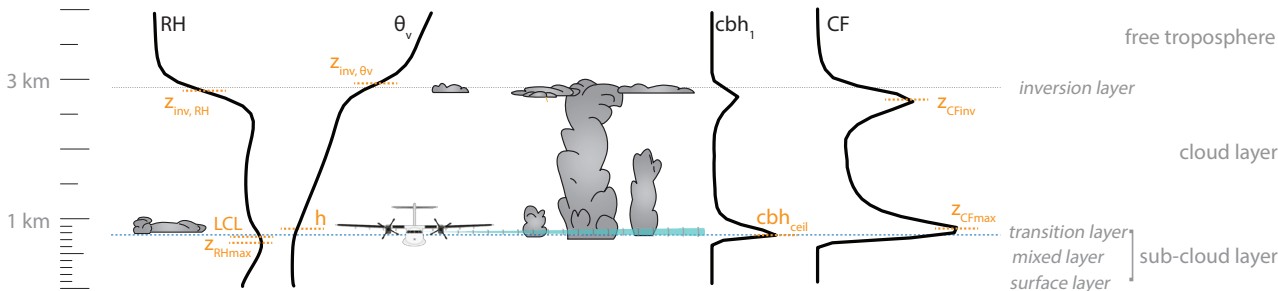

**Figure 3.** Schematic representation of the trade-wind layer with the different levels considered as part of the ground support to determine the cloud-base level of the ATR, plus sometimes the cloud-top level. The subcloud layer top, referred to as $h$ or $z_{SC}$, is defined as the level of neutral buoyancy of a parcel originating from the surface layer with a 0.2 K excess in $\theta_v$ (section 2.5); $z_{CFmax}$ is the level of maximum near-base cloud fraction (as would be seen for instance in ground-based radar observations), $cbh_{ceil}$ is the peak of the distribution of the first-detected cbh of the ceilometer, $z_{RHmax}$ is the level of maximum relative humidity (RH) at the mixed layer top, and LCL is the lifting condensation level. $z_{inv,RH}$ and $z_{inv,\theta_v}$ are the inversion heights based on the maximum gradients in RH and $\theta_v$, and $z_{CFinv}$ is the second level of maximum cloud fraction around the inversion.

(BCO, Stevens et al. (2016)) and R/V Meteor, as well as satellite imagery from the GOES-16 Advanced Baseline Imager (https://doi.org/10.7289/V5BV7DSR).

During the flights, real-time ATR lidar backscatter quick-looks and visual impressions from the pilots, as well as real-time information from the HALO dropsondes and lidar (WALES) quick-looks (Konow et al., 2021) were used to fine-tune the flight level. To provide spatial context between the East-West anchor points (R/V Meteor on the eastern side and BCO on the western

side), satellite imagery and HALO data were used to anticipate horizontal gradients in the levels. In case the cloudiness was associated with very shallow clouds and the cloud-base height was exhibiting strong gradients across the sampling area, a slight adjustment in the cloud-base flight level along the rectangle or in between subsequent rectangles was allowed to improve the sampling of clouds. Occasionally, the cloud-base level was slightly adjusted between the northern and southern halves of a given rectangle. However it was never adjusted between the eastern and western sides of the rectangle, so that the cloud field

within the rectangle was sampled at the same height by the horizontal lidar-radar measurements performed from opposite sides of the rectangle (see Chazette et al. (2020) for an illustration of the sampling by horizontal lidar measurements).

At the beginning of the last rectangle of each flight, the level of the L-patterns to be flown within the sub-cloud layer was determined. The first L-pattern was flown near the top of the subcloud layer, about 150-200 m below the lowest cloud base leg (to make sure no cloud is present), and the second L-pattern was flown near the middle of the subcloud-layer. Finally, shortly

before the ferry back to Barbados and when daylight was still present, the ATR flew short straight legs near the sea surface (S-pattern).

Over the campaign, the cloud base flight level ranged from about 600 to 850 m, the L-pattern near the top and the middle of the subcloud-layer were flown around 500-600 m and 200-400 m, respectively, and S-patterns were flown about 60 m above the sea surface (Table 1, Fig. 4).



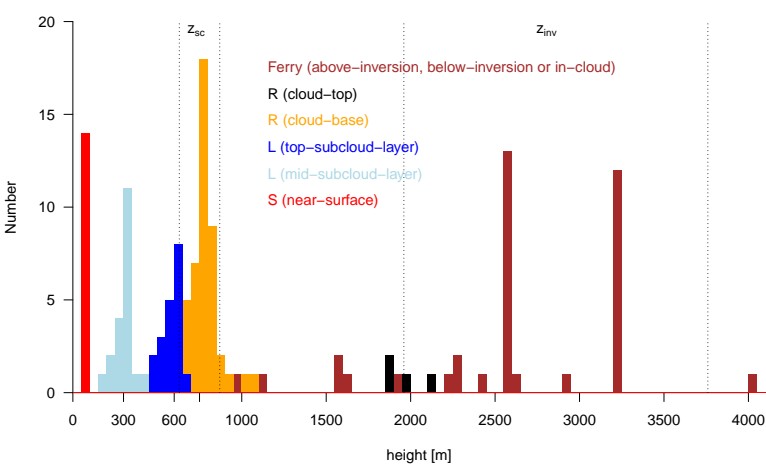

**Figure 4.** Vertical distribution of the number of R-, L- S- and Ferry patterns flown by the ATR during the whole EUREC[4]A campaign. Also reported are the ranges of subcloud-layer top heights ($z_{sc}$) and inversion heights ($z_{inv}$) derived from dropsondes (Table 3).

### 2.4 Flight segmentation

To aid in the analysis of the flight data, each flight is segmented into non-exclusive timestamps summarized in a set of YAML files (Table 2). Different kinds of segments are defined, that correspond to basic patterns ('R-pattern', 'L-pattern', 'S-pattern') or to particular phases of the flight (e.g., 'Ferry'). The vertical level at which these patterns are flown (at cloud-top, cloud-base, near the top of the subcloud-layer, near the middle of the subcloud-layer, near the sea surface, above or below the trade inversion level) is also indicated as a 'note' in the YAML files. The vertical excursions of the ATR are referred to as 'Profiles', and the direction (upward or downward) in which they were realized is also reported. An example of flight segmentation is shown for RF11 (Fig. 5). The vertical and horizontal trajectories of each flight are shown in Figs. B1 and B2.

The characterization of the turbulence ("T") requires to consider straight and stabilized legs of at least 30 km (Lenschow et al., 1994). For this reason, the R- and L-patterns were also associated with a finer segmentation in straight horizontal legs of equal duration and length (Fig. 6 from Brilouet et al. (2021)): short segments of approximately 30 km (5 min flight) are referred to as 'T-shortlegs', and longer segments of approximately 60 km are referred to as 'T-longlegs'. The longest stabilized segments in one direction are also reported as 'T-longestlegs'; in contrast with the 'T-shortlegs' or 'T-longlegs', these segments can have various lengths, ranging from 60 to 125 km.



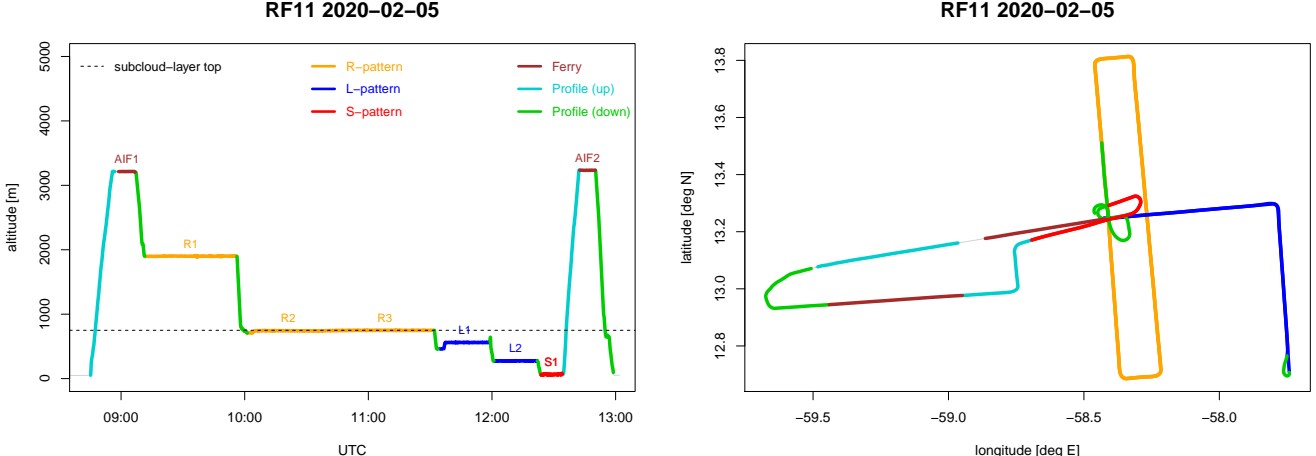

**Figure 5.** (a) Time-height trajectory and (b) longitude-latitude trajectory of the ATR during RF11 (Feb 5th, 2020), illustrating the different patterns and segments of the flight. Also reported is the subcloud-layer top diagnosed from HALO dropsondes (Table 3).

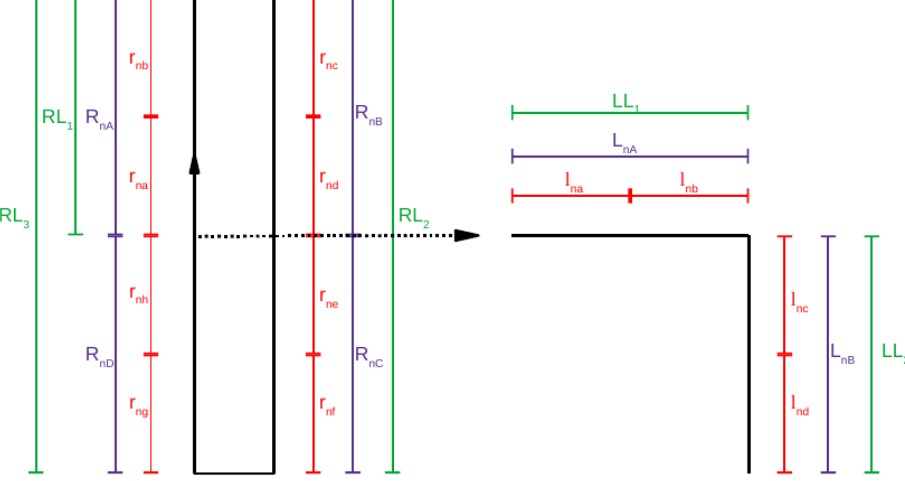

**Figure 6.** Segmentation of the R- and L-patterns into straight and stabilized segments of equal duration and length for turbulence studies (T-shortlegs: 30 km/5 min in red, referred to as $r_{nx}$ or $l_{nx}$ where $n$ is the pattern number, T-longlegs: 60 km/10 min in purple, referred to as $R_{nX}$ or $L_{nX}$). Also reported are the longest stabilized legs in one direction (T-longestlegs, 120 km/20 min or 60 km/10 min, in green, referred to as $RL_i$ or $LL_i$ where $i = 1,..P$ where P is the number of such segments for the flight). A similar nomenclature is used for the segmentation of the S-patterns. See Table 2 for the definition and the nomenclature of these segments. After Brilouet et al. (2021).



| kind | note | name | geometry | number |
|---|---|---|---|---|
| R-pattern | cloud-top | $R_n$ | rectangle of 120 km × 20 km | 4 |
| | cloud-base | | | 50 |
| L-pattern | top-subcloud-layer | $L_n$ | 2 perpendicular legs of 60 km (10 min) | 18 |
| | mid-subcloud-layer | | | 22 |
| S-pattern | near-surface | $S_n$ | 40 km (7 min) | 14 |
| Ferry | above-inversion ferry leg | $AIF_n$ | | 35 |
| | below-inversion ferry leg | $BIF_n$ | | 7 |
| | in-cloud ferry leg | $ICF_n$ | | 3 |
| Profile | upward | $UP_n$ | | 50 |
| | downward | $DN_n$ | | 95 |
| T-shortlegs | cloud-top | $r_{nx}$ | 30 km (5 min) | 31 |
| | cloud-base | | | 354 |
| | top subcloud-layer | $l_{nx}$ | | 63 |
| | mid subcloud-layer | | | 65 |
| | near-surface | $s_{nx}$ | | 22 |
| T-longlegs | cloud-top | $R_{nX}$ | 60 km (10 min) | 16 |
| | cloud-base | | | 182 |
| | top subcloud-layer | $L_{nX}$ | | 33 |
| | mid subcloud-layer | | | 32 |
| | near-surface | $S_{nX}$ | | 16 |
| T-longestlegs | cloud-top | $RL_n$ | 120 km (20 min) or 60 km (10 min) | 15 |
| | cloud-base | | | 118 |
| | top subcloud-layer | $LL_n$ | | N |
| | mid subcloud-layer | | | N |
| | near-surface | $SL_n$ | | 16 |

**Table 2.** Segmentation of the ATR flights into patterns ('kind'), flown at different levels ('note'). Each segment is associated with a 'name', where n = 1, 2,.. N (N being the number of patterns of the 'kind' category flown during the flight), X = A, B, C.... and x = a, b,.. h. See Fig. 6 for an illustration of the sub-segmentation of the patterns into T-shortlegs, T-longlegs and T-longestlegs segments. Also reported is the total number of segments in each category. This information is included in a set of YAML files (one file per flight).

## 2.5 Environmental conditions associated with each flight

To aid in the analysis of the ATR data, we summarize in Tables 3 and 4 the main environmental conditions associated with each flight, as well as qualitative descriptions of the prominent cloud types and mesoscale cloud patterns present during each flight, plus some information about aerosols and the presence of precipitation. The prominent cloud types are determined by





| Flight | Date | $z_{LCL}$ | $z_{SC}$ | $z_{INV}$ | $\omega_{SC}$ | $\omega_{INV}$ | $\omega_{FT}$ | $RH_{FT}$ | PW | EIS | LTS | $V_s$ |
|---|---|---|---|---|---|---|---|---|---|---|---|---|
| | | m | m | m | hPa day$^{-1}$ | hPa day$^{-1}$ | hPa day$^{-1}$ | % | mm | K | K | m s$^{-1}$ |
| RF03 | 01-26 | 683 | 700 | 2060 | -24.46 | 30.63 | -38.32 | 4.71 | 28.93 | -0.34 | 15.79 | 4.88 |
| RF04 | 01-26 | 727 | 690 | 2800 | 44.99 | 5.56 | -42.52 | 4.81 | 29.64 | -1.22 | 14.65 | 3.08 |
| RF05 | 01-28 | 503 | 630 | 2650 | 32.86 | 21.71 | 40.33 | 12.69 | 40.17 | -1.60 | 15.99 | 7.48 |
| RF06 | 01-30 | 570 | 650 | 2670 | NA | NA | NA | 9.45 | 36.39 | -2.47 | 14.36 | 8.61 |
| RF07 | 01-31 | 534 | 640 | 2440 | -4.45 | 128.82 | -47.21 | 6.24 | 39.79 | -3.06 | 14.37 | 7.23 |
| RF08 | 01-31 | 525 | 660 | 3490 | 6.25 | 55.93 | 58.36 | 6.51 | 38.11 | -3.28 | 14.27 | 6.43 |
| RF09 | 02-02 | 719 | 730 | 2760 | 18.87 | -29.50 | 37.28 | 5.70 | 34.68 | 2.06 | 17.85 | 6.34 |
| RF10 | 02-02 | 681 | 700 | 2910 | 32.70 | 29.52 | 22.50 | 6.00 | 34.90 | 0.66 | 16.83 | 5.39 |
| RF11 | 02-05 | 565 | 750 | 2850 | -65.94 | 71.06 | 96.74 | 6.59 | 36.71 | -1.52 | 15.26 | 10.01 |
| RF12 | 02-05 | 589 | 730 | 2930 | -10.35 | 31.92 | 61.23 | 6.98 | 36.28 | -2.07 | 14.71 | 9.40 |
| RF13 | 02-07 | 758 | 790 | 2850 | 95.51 | -12.03 | -12.28 | 8.74 | 32.62 | 0.53 | 15.96 | 11.70 |
| RF14 | 02-07 | 747 | 870 | 2810 | 47.77 | 27.28 | 84.52 | 9.75 | 34.11 | 0.21 | 15.90 | 9.55 |
| RF15 | 02-09 | 605 | 770 | 2770 | -59.37 | 109.04 | 69.89 | 16.89 | 34.76 | -3.65 | 12.87 | 10.34 |
| RF16 | 02-09 | 644 | 820 | 3050 | -32.78 | 99.56 | 108.33 | 15.12 | 36.72 | -3.56 | 12.81 | 11.05 |
| RF17 | 02-11 | 619 | 730 | 3760 | -17.89 | -192.49 | -137.04 | 21.07 | 42.51 | -3.51 | 12.88 | 10.96 |
| RF18 | 02-11 | 605 | 760 | 2860 | 23.10 | -42.02 | -172.31 | 26.12 | 43.72 | -3.35 | 13.33 | 10.32 |
| RF19 | 02-13 | 622 | 740 | 1960 | 81.38 | 4.42 | 84.52 | 29.29 | 44.79 | -2.47 | 13.86 | 10.87 |
| RF20 | 02-13 | 68 | 740 | 2110 | 12.33 | -33.00 | 111.24 | 31.40 | 42.74 | -2.58 | 13.57 | 10.72 |
| average | | 632 | 727 | 2762 | 10.62 | 18.02 | 19.13 | 12.67 | 37.09 | -1.73 | 14.74 | 8.58 |
| std dev | | 75 | 61 | 432 | 42.95 | 71.15 | 80.89 | 8.51 | 4.41 | 1.67 | 1.40 | 2.45 |

**Table 3.** Meteorological conditions associated with each ATR flight, and their average over all flights. All quantities are computed from the JOANNE dropsondes dataset (George et al., 2021) as averages over 3 consecutive circles flown during each ATR flight. $z_{INV}$, $z_{SC}$ and $z_{LCL}$ are the trade-inversion height, the subcloud-layer top height and the lifting condensation level height, respectively. $z_{INV}$ is defined as the height where the moist static energy is minimum between 1300 and 4000 m. $z_{SC}$ is defined as the lowest altitude above 200 m where $\theta_v(z)$ exceeds by more than 0.2 K the mass-weighted average of $\theta_v$ from 200 m to z (Canut et al., 2012; Rochetin et al., 2021; Touzé-Peiffer et al., 2021). $z_{LCL}$ is diagnosed as $z_{LCL} = z_{20m} - (C_{pd}((T_{LCL}-T_{20m})/g))$, with $T_{LCL} = 1/((1/(T_{20m}-55)) - (\log(RH_{20m})/2840)) + 55$ where $T$ is the temperature and RH the relative humidity. $\omega$ is the vertical velocity measured at the scale of the EUREC$^4$A circle by dropsondes (Bony and Stevens, 2019; George et al., 2021); $\omega_{SC}$ and $\omega_{INV}$ are the mass-weighted averages of $\omega$ in a 200 m layer centered around $z_{SC}$ and $z_{INV}$, respectively. $\omega_{FT}$ and $RH_{FT}$ (FT referring to the lower free troposphere), are the mass-weighted averages between 4000 and 6000 m of $\omega$ and RH, respectively (note that $\omega$ was not measured during RF06). PW (precipitable water) is the mass-weighted integral of water vapor specific humidity from the surface to the altitude of the dropsonde launch (about 10 km). EIS (the estimated inversion strength) and LTS (the lower-tropospheric stability) are two measures of the lower-tropospheric stability defined as LTS = $\theta_{700hPa} - \theta_{1000hPa}$ (Klein and Hartmann, 1993) and EIS = LTS - $\Gamma_m^{850}(z_{700hPa} - z_{LCL})$, where $\Gamma_m^{850}$ is the moist-adiabatic $\theta$ gradient at 850 hPa (Wood and Bretherton, 2006). Note that the surface relative humidity used in this calculation is derived from each dropsonde at 20 m rather than assumed to be fixed and equal to 80% as in Wood and Bretherton (2006). $V_s$ is the near-surface wind speed computed from the zonal and meridional wind components measured by dropsondes at 20 m.





| Flight | Date | Low-level clouds | | Aerosols | | | Precipitation | |
| | | cloud types | cloud pattern | AEC | VDR | dust | drizzle | rain |
| | | (VS, CH, StCH, ExStCH, CS) | (SU, GR, FL, FI) | km$^{-1}$ | % | | % | % |
|---|---|---|---|---|---|---|---|---|
| RF03 | 01-26 | VS, CH, StCH | **FI, GR** | - | - | | 4.3 | 2.2 |
| RF04 | 01-26 | StCH, CS, VS, CH | **FI, GR** | 0.02±0.02 | 0.8±0.1 | | 0.6 | 0.2 |
| RF05 | 01-28 | VS, CH | SU, GR | 0.06±0.04 | 0.5±0.1 | | 0.1 | 0 |
| RF06 | 01-30 | VS | **SU** | 0.09±0.10 | 1.4±0.5 | + | 0.1 | 0 |
| RF07 | 01-31 | StCH, CH, VS | SU, GR, FL | 0.14±0.06 | 2.1±0.2 | ++ | 2.5 | 1.3 |
| RF08 | 01-31 | StCH, VS, CH, ExStCH | SU, GR, FL | 0.2±0.08 | 2.2±0.3 | ++ | 1.2 | 0.6 |
| RF09 | 02-02 | CS, CH, StCH | **FL** | 0.14±0.06 | 3.0±0.6 | ++ | 0 | 0 |
| RF10 | 02-02 | CH, StCH, ExStCH | **FL** | 0.16±0.04 | 2.7±0.4 | ++ | 1.1 | 0.2 |
| RF11 | 02-05 | CH, StCH | GR, FL | 0.13±0.08 | 1.4±0.1 | + | 1.9 | 0.2 |
| RF12 | 02-05 | VS, CH | GR, SU | 0.13±0.07 | 1.4±0.2 | + | 0.2 | 0 |
| RF13 | 02-07 | VS, CH, StCH | **FL**, FI | 0.06±0.04 | 0.4±0.3 | | 0.1 | 0 |
| RF14 | 02-07 | VS, CH, StCH | **FL**, FI | 0.04±0.04 | 0.3±0.2 | | 0.3 | 0 |
| RF15 | 02-09 | VS, CH, StCH | **SU, GR** | 0.18±0.10 | 0.6±0.1 | | 0.3 | 0.1 |
| RF16 | 02-09 | VS, CH, StCH | **SU, GR** | 0.18±0.07 | 0.9±0.2 | | 2.3 | 0.8 |
| RF17* | 02-11 | ExStCH | **FL**, SU | 0.15±0.16 | 0.7±0.1 | | 9.1 | 9.8 |
| RF18* | 02-11 | ExStCH, VS | **FL**, SU | 0.19±0.13 | 1.0±0.2 | + | 5.1 | 9.3 |
| RF19 | 02-13 | StCH, CS | GR, FL, FI | 0.09±0.08 | 0.6±0.3 | | 3.4 | 1.3 |
| RF20* | 02-13 | CS, VS, ExStCH | GR, FL, FI | 0.05±0.04 | 0.6±0.4 | | 1.6 | 0.6 |

**Table 4.** Cloud, aerosol and precipitation conditions associated with ATR flights. Through the combined analysis of Fig. B2, GOES-E animations (section A), BCO radar information and C³ONTEXT results (Schulz, 2021), the prominent low-level cloud types (at the scale of the R- and L-patterns) and cloud mesoscale patterns (at the scale of the EUREC⁴A circle) are reported for each ATR flight. The different low-level cloud types considered are very shallow cumuli (VS), vertically developped chimney clouds (CH), chimney clouds with stratiform outflow below the inversion (StCH) and chimney clouds with an horizontally extended stratiform layer (ExStCH). Clear-sky is referred to as CS. The mesoscale cloud patterns (referred to as SU, GR, FL or FI for Sugar, Gravel, Flowers and Fish) are defined in Stevens et al. (2020). They are written in bold when there is a consensus about their prominence during the flight. The aerosol extinction coefficient (AEC), volume depolarization ratio (VDR) and dust condition are from Chazette et al. (2020); *dust+* corresponds to 1%≤ VDR < 2% and *dust++* to VDR ≥ 2%. The fractional areas (in %) of the R-patterns flown at cloud-base covered by drizzle or rain are derived from the BASTA radar using reflectivity thresholds of -20 dBZ and 0 dBZ to distinguish clouds from drizzle and drizzle from rain, respectively (section 3.5.3). Asterisks * indicate the presence of deeper congestus clouds with cloud-top at 5 km (for RF17 and RF18) or alto-stratus layers between 5 and 8 km for RF20.





watching animations of the GOES-16 satellite imagery centered on each ATR flight (see their description in Appendix A) plus
BCO radar observations. The prominent mesoscale cloud patterns are determined visually from the analysis of the GOES-16
movies associated with each ATR flight and the results of the mesoscale cloud patterns overview of Schulz (2021).

Daily reanalyses from ERA5 (Hersbach et al., 2020) suggest that over the EUREC$^4$A circle, the sea surface temperature
(which corresponds to the foundation temperature and is free from diurnal variations) was 26.9 °C on average and exhibited
day-to-day variations of only ± 0.1 °C. On the other hand, weather conditions varied considerably during the campaign (Ta-
ble 3): the first day of ATR operations (Jan 26) was associated with much drier conditions in the free troposphere and much
weaker trade winds than the last day of operation (Feb 13); the lower-tropospheric stability was particularly high during RF09-
10 (Feb 2) and RF13-14 (Feb 7), and particularly low during RF15-16 (Feb 9) and RF17-18 (Feb 11); $\omega$ in the lower free
troposphere was associated with a large-scale ascent during RF17-18 (Feb 11) but it was associated with subsidence on RF05
(Jan 28), RF11-12 (Feb 5) and RF15-16 (Feb 9); the LCL and subcloud heights were particularly low on RF07-08 (Jan 31) and
particularly high on RF13-14 (Feb 7).

Consistently with these contrasted environmental conditions, the most prominent cloud types and mesoscale cloud patterns
encountered during each flight also varied (Table 4). For instance, small thin clouds prevailed during RF05 and RF06 (Jan
28 and Jan 30), but deeper cloud systems associated with the presence of stratiform cloudiness around the trade inversion
level and rain were present during RF03 (Jan 26), RF07 (Jan 31), RF17-18 (Feb 11) and RF19 (Feb 13). The mesoscale cloud
patterns associated with each ATR flight were often a mix of several patterns. Yet, a few flights were associated with a greater
prominence of specific mesoscale patterns. For instance, RF06 (Jan 30) was clearly associated with a *Sugar* pattern, while RF09
and RF10 (on Feb 2) were clearly associated with a *Flowers* pattern, RF09 sampling mostly the clear-sky part of the pattern
and RF10 sampling more of the cloudy area. The *Gravel* pattern occurs often in association with other patterns, especially with
the *Sugar* pattern, as found during RF05 (Jan 28), RF12 (Feb 5), RF15 and RF16 (Feb 9).
Finally, an episode of dust occurred from Jan 31 to Feb 5, and on Feb 11, as also observed from Ragged Point in Barbados
and from the R/V Ron Brown (Stevens et al., 2021).

## 3   Instrumentation and datasets

The ATR instrumentation used for EUREC$^4$A (Fig. 7) was composed of an ensemble of in-situ probes and sensors to measure
the dynamical, thermodynamical and microphysical properties of the atmosphere near the aircraft, passive radiometers to
measure broadband radiative fluxes and spectrally-resolved infrared radiances, a laser spectrometer to measure the isotopic
composition of water vapor in-situ, and a lidar and two Doppler cloud radars to characterize the macrophysical properties of
clouds and the presence of precipitation and aerosols away from the aircraft. All instruments are used in the EUREC$^4$A datasets
presented in this paper except the Gerber, Nevzorov, FSSP300 and FCDP probes.

The quality control, the calibration and the processing of the datasets derived from the core instrumentation of the ATR
(referred to as SAFIRE-CORE, SAFIRE-RADIATION, SAFIRE-CLIMAT and SAFIRE-CAMERA), from the microphysical
probes (UHSAS and PMA), from the Doppler cloud radars (BASTA and RASTA) and from the combined radar-lidar dataset



**Figure 7.** Location on the ATR of the main instruments discussed in this paper. Upper to lower panels show the aircraft from different view points: right, left, bottom and top, respectively. The exact positions of each instrument are given in Tables 5 to 9. Note that the Gerber, Nevzorov, FSSP300 and FCDP probes are not used in the EUREC[4]A datasets presented in this paper.



(BASTALIAS) are presented below. The processing of the lidar dataset (ALIAS), the turbulence dataset (SAFIRE-TURB) and the isotopic dataset (Picarro) are fully described in separate papers (Chazette et al. (2020), Brilouet et al. (2021) and Bailey et al. (submitted), respectively); only the main aspects of these datasets are summarized below.

## 3.1 Aircraft navigation, attitude and meteorological data (SAFIRE-CORE)

### 3.1.1 Inertial/Navigation system

The ATR Inertial Navigation System (INS), also named AIRINS, is an iXblue inertial navigation system using a Fiber-Optic Gyroscope. By construction, an inertial unit is drifting and the position needs to be reset by a GPS position to provide accurate parameters. It is done by using a Trimble BX992 GPS. The AIRINS-GPS positioning system then provides groundspeed, acceleration, attitudes angles and speed platform components in an Earth-based coordinate system.

During EUREC⁴A, three problems occurred that impacted the measurements and the data processing: (1) A failure in the internet ouput of the AIRINS-GPS system prevented us from recording the data at 100 Hz as usual; the data were recorded instead at 50 Hz on a serial output, and then they were synchronized and averaged at 25 Hz and at 1 Hz. (2) During RF03, the GPS was rejected by AIRINS, which resulted in an incorrect position (true heading and attitude) and thus unreliable horizontal wind measurements for this flight; a corrected position (derived from the GPS only) was used in the V2 version of the SAFIRE-CORE dataset, as well as in the RF03 files of other ATR datasets. (3) For RF20, the inertial/GPS data are available at 1 Hz only.

### 3.1.2 Pressure, anemoclinometric and wind measurements

The SAFIRE ATR is equipped with a five-hole radome that measures the distribution of pressure around the nose of the aircraft (Table 5): the difference of pressure measured between two holes in the vertical or horizontal planes informs about the attack angle and sideslip angle, respectively (Lenschow, 1986). The static and dynamic pressures are measured by Rosemount or Thales transducers connected to Pitot tubes on both sides of the radome. The static pressure, which corresponds to the pressure corrected from the airflow disturbance produced by the aircraft, is determined using a pre-established calibration based on specific flights and maneuvers. The dynamical pressure is obtained by subtracting the static pressure from the total pressure measured at the central radome hole. The true air speed (TAS), which is the speed of the aircraft relative to the airmass through which it is flying, is calculated from the dynamical and static pressures.

The wind is then inferred from the difference between the speed of the aircraft relative to the Earth and the true air speed (Lenschow, 1986). The high rate wind measurements of the ATR have been very robust since its first field campaign in 2006 (Saïd et al., 2010). Unfortunately, because of a hose leak between a hole of the radome and a pressure transducer inside the radome, the measurement of the vertical wind is not reliable from RF02 to RF08. The horizontal wind measurements were not significantly affected by this problem.



| Instrument | Brief description | Position on ATR | |
|---|---|---|---|
| 5-hole radome | For measuring the differential pressure around the nose of the aircraft | radome | |
| Pitot probes | Rosemount and Thales transducers connected to Pitot probes measuring static and dynamic pressure | fuselage | |
| Rosemount 1 | E102AL non-deiced temperature sensor | nose (right-hand side); [N1-4, FR2-3] | |
| Rosemount 2 | E102AL deiced temperature sensor | fuselage (right-hand side); [N1-8, FR15-16] | |
| Fine wire | fine wire resistance for measuring fast temperature fluctuations | nose (left-hand side); [N1-1, FR2-3] | |

**Table 5.** Core instrumentation of the ATR for pressure and temperature measurements. See Annex C for the correspondance between the position H, N or FR and the ATR configuration (H refers to an aircraft window, N to the nose of the aircraft and FR to a particular position along the fuselage).

### 3.1.3 Air temperature

During EUREC$^4$A, the air temperature was measured by 2 Rosemount sensors E102AL (Table 5). The first one is located on the nose of the aircraft, inside a non-deiced housing, and the second one is located on the fuselage inside a deiced housing (Fig. 7). The static temperature, which is the temperature corrected for aircraft speed and recovery factor of the housing, is calculated as:

$$T_s = \frac{T_t}{1 + r_f \left( \left( 1 + \frac{\Delta P}{P_s} \right)^{R_a/c_{pa}} - 1 \right)} \qquad \text{if } \Delta P > 6$$

where $T_t$ is the measured total temperature (°C), $\Delta P$ the dynamic pressure (hPa), $P_s$ the static pressure (hPa) and $r_f$ the recovery factor ($r_f$=0.98)





From RF09 to RF20, fast (turbulent) temperature fluctuations were also measured at 200 Hz (and averaged at 25 Hz) with a fine wire temperature sensor. The fine wire is a 5 $\mu$m platinium wire soldered on a support and mounted inside a SFIM T4113 housing. Despite its fragility (a fine wire can easily break during takeoff or landing when the aircraft encounters particles or insects), it remained intact during the whole campaign. Despite its housing, the response time of the Rosemount sensor can sometimes be affected by the presence of cloud droplets. The fine wire can also be affected by this problem, but it recovers much more quickly, emphasizing the complementarity of the two sensors (Brilouet et al., 2021). The total temperature from the fine wire is derived by fitting and calibrating its raw measurements against the total temperature measured by the non-deiced Rosemount sensor. The resistance of the fine wire being subject to oxidation, this calibration is performed for each individual flight. The static temperature is estimated using the same method as for the Rosemount sensor, using (for the lack of better estimate) the same recovery factor.

The Rosemount and fine wire temperature data are processed at 1 Hz and at 25 Hz. From RF09 to RF20, the turbulence dataset (SAFIRE-TURB) uses the fine wire data as the best estimate for fast fluctuations, and the Rosemount data as a spare (Brilouet et al., 2021).

### 3.1.4  Humidity

No less than five instruments measured humidity in-situ on board the ATR (Table 6), in addition to the cavity ring-down spectrometer (CRDS) presented in another section of this paper (section 3.6). Each instrument is based on a particular measurement principle or technology, and therefore exhibits specific strengths and limitations in terms of stability, response rate, sensitivity to the presence of condensation or measuring range. The comparison and fine analysis of the different measurements makes it possible to calibrate and correct or bypass the shortcomings of each measurement, so as to produce high quality humidity datasets. The main features associated with these instruments and the processing of their measurements are outlined below.

A chilled mirror dew point hygrometer (Buckresearch 1011C) measured the atmospheric dew and frost points. This measurement, made by cooling a reflective condensation surface until an optical system detects the presence of condensation, is traditionally considered as a reference measurement for humidity. However, this type of hygrometer can have limitations when the aircraft undergoes large changes in altitude, passes through a cloud or samples environments with high humidity contrasts. This sensor also has a slow response time and show limitations in very dry conditions such as those encountered above the trade inversion.

A Humicap 180C Enviscope-Vaisala capacitive sensor was placed inside a non-deiced Rosemount E102 housing. This sensor is made of a hygroscopic dielectric material whose capacitance is dependent on humidity. After correcting for the effects of aircraft speed, it measures relative humidity directly with a short response time. However, the sensor is sensitive to the presence of cloud droplets and it can report relative humidities above 100 %. Its measurements are thus considered only in unsaturated environments, and under these conditions they help assess the robustness or even calibrate the measurements of other sensors.

Unlike previous sensors, the Water Vapor Sensing System (WVSS-II) from SpectraSensors can measure humidity with a good reliability and regularity, without being affected by the presence of cloud droplets or very dry air. This is due to its particular technology, based on tunable diode laser absorption spectroscopy in the near-infrared (1.37 $\mu$m), and to the fact that



| Instrument | Brief description | Position on ATR | |
|---|---|---|---|
| Dew point | Buckresearch 1011C chilled mirror dew point hygrometer | window (right-hand side); [H32, FR32-33] | |
| WVSSII | Water Vapor Spectro Sensor; near-infrared tunable diode laser absorption | window (right-hand-side); [H6, FR19-20] | |
| HUMAERO | Enviscope hydroscopic dielectric material placed between a pair of electrodes (capacitive hygrometer) | window (right-hand-side); [N1-10, FR15-16] | |
| Picarro inlet | L2130-i cavity ring-down laser spectrometer | window (right-hand-side); [inlet: H6, FR19-20] | |
| Licor 7500A | Near-infrared gas analyzer for measuring rapid humidity fluctuations | window (left-hand-side); [H3, FR19-20] | |
| KH20 | Campbell krypton hygrometer for measuring rapid humidity fluctuations | nose (left-hand-side); [N1-3, FR3-4] | |

**Table 6.** Humidity sensors. Note that the cavity ring-down spectrometer (whose inlet is shown here) is represented in Table 9. See Annex C for the correspondance between the position on the aircraft and the ATR configuration.



its sampler has been designed to minimise the biases associated with the presence of cloud droplets or aerosols. Therefore, for this campaign it is considered as a reference for slow humidity measurements, and it is used to adjust or calibrate humidity measurements from other sensors. The WVSS-II measures the mixing ratio of water vapor relative to dry air in ppmv. The volumic concentration is converted to a mass concentration to provide absolute humidity measurements in $g\,m^{-3}$.

Finally, two additional instruments were used to measure rapid fluctuations of humidity: a Licor LI-7500A and a Campbell Scientific krypton hygrometer (KH20).

The Licor LI-7500A is a near-infrared gas analyser originally designed to measure eddy-covariance fluxes on ground towers, which has been adapted by SAFIRE to perform airborne measurements. Its strength lies in its short time response, but its main limitation is its high sensitivity to the presence of liquid water (its performance can be affected even a few seconds after leaving a cloud). Periods when the humidity measurement is affected by condensation (typically inside clouds) are detected on the basis of the strength of $CO_2$ measurements made by the same sensor. The Licor performance can also be affected by the presence of sea salt, particularly when the aircraft is flying low near the sea surface. To minimise this problem, the lowest legs were performed at the end of the flight, and the Licor window was cleaned before each subsequent flight. The Licor humidity measurements (in $g\,m^{-3}$) are calibrated against the WVSS-II absolute humidity measurements of RF13, and the calibration coefficients are the same for all flights. As the Licor clock is initialized manually, it is sometimes delayed by a few seconds. This delay is subsequently corrected during post-processing. The corrected and synchronized time parameter of the Licor instrument is also used to correct a delay of 3 s of the WVSS-II sensor induced by the interface of the instrument. Note that Licor data were not recorded during the flights RF05 and RF06.

The KH20 uses the absorption of the UV light emitted at 123.58 and 116.49 nm by a krypton lamp to estimate the water vapor density. Also originally designed for eddy-covariance measurements on ground towers, this instrument has been heavily modified by SAFIRE to be operated on the ATR: the housing of an older humidity sensor (a Lyman-alpha hygrometer) was used to install the source lamp and detector, and the electronic box was installed inside the cabin. This sensor was less sensitive to the presence of cloud droplets than the Licor, but it was more affected by sea salt. Therefore, as the Licor it was cleaned before each subsequent flight. The KH20 measures rapid fluctuations of humidity but not absolute humidity. Absolute values (in $g\,m-3$) are obtained by calibration against the slow (1 Hz) humidity measurements of the WVSS-II (Brilouet et al., 2021).

Based on the processing of these different measurements, two humidity datasets have been produced: one at 1 Hz, included in the SAFIRE-CORE dataset, and another at 25 Hz, which is included in the SAFIRE-TURB dataset. Note that in the SAFIRE-TURB dataset, the calibration of the humidity measurements is performed on a leg by leg basis, both for the Licor 7500A and the KH20 sensors.

## 3.2 Radiative measurements

### 3.2.1 Broadband radiative fluxes (SAFIRE-RADIATION)

Kipp and Zonen sensors mounted at the top and at the bottom of the ATR measured upwelling and downwelling broadband radiative fluxes (Table 7): CGR4 pyrgeometers measured hemispheric longwave fluxes in the 4.5–42 $\mu$m spectral range,



| Instrument | Brief description | Position on ATR | |
| --- | --- | --- | --- |
| CLIMAT CE332 | Downward-staring infrared radiometer measuring irradiance at 8.7 $\mu$m, 10.8 $\mu$m and 12.0 $\mu$m and used to infer SST | bottom; [OB1, FR15-16] | |
| Pyrgeometer | Kipp and Zonen CGR4 sensor measuring hemispheric broadband upwelling and downwelling longwave (4.5–42 $\mu$m) | bottom and top; [EB, FR22-23 and EH, FR23-24] | |
| Pyranometer | Kipp and Zonen CMP21 and CMP22 sensors measuring hemispheric upwelling and downwelling shortwave in two spectral ranges: 0.75–2.7 $\mu$m (red dome) and 0.2–3.6 $\mu$m (clear dome) | bottom and top; [EB, FR22-23 and EH, FR23-24] | |
| Cameras | High-resolution visible cameras (AV GT1920C and Mako G-223) looking sideways and downward (respectively) | bottom [OB3, FR15-16] and right-hand side window [H4, FR18-19] | |

**Table 7.** Core instrumentation of the ATR for radiative measurements. See Annex C for the correspondance between the position on the aircraft and the ATR configuration.

CMP21 pyranometers measured hemispheric shortwave radiation in the 0.75–2.7 $\mu$m spectral range (red dome), and CMP22 pyranometers measured hemispheric shortwave radiation in the 0.2–3.6 $\mu$m spectral range (clear dome).

Measuring upwelling and downwelling radiative fluxes requires the aircraft to be in a plane and stable position. For this
reason, the SAFIRE-RADIATION dataset includes two sets of variables for each radiative flux: raw fluxes, and fluxes corrected for the attitude of the aircraft. In the time series of corrected fluxes, whenever the roll or pitch of the aircraft was greater than $\pm$ 5° the radiative measurements were considered as 'undefined', and otherwise the downwelling shortwave measurements were corrected for the attitude of the aircraft. This correction requires to know the offset of the sensor installation, which corresponds to the bias associated with the potential tilt of the mechanical installation of the sensors relative to their support.
This offset must be estimated every time the sensor has been re-mounted on the aircraft (such as done at the arrival of the ATR in Barbados, section 2.1). It was determined through specific manouvers performed during the test flight RF02.



All pyrgeometers and pyranometers worked properly during the campaign except one: the CMP21 pyranometer (red dome) at the top of the aircraft. Because of this malfunctioning, the downwelling 0.75–2.7 $\mu$m irradiance measurements were either absent or unvalidated during the campaign. However all other upward and downward longwave and shortwave fluxes, including

the downwelling shortwave measurements over the 0.2–3.6 $\mu$m spectral range, are available and distributed in the SAFIRE-RADIATION dataset at 1 Hz.

### 3.2.2 Infrared brightness temperatures (SAFIRE-CLIMAT)

In addition to broadband radiometers, the ATR carried a nadir-viewing multispectral radiometer, the CLIMAT CE332 instrument, developed by the Laboratoire d'Optique Atmosphérique (LOA) in collaboration with CIMEL (Brogniez et al., 2003).

This radiometer measures infrared radiances and brightness temperatures at three wavelengths: 8.7, 10.6 and 12 $\mu$m (Table 7). It is done by comparing the radiances measured on the observed target with that measured by looking at a reference cavity maintained at a given temperature. During the post-processing, the measurements performed at 6 Hz are synchronized and averaged at 1 Hz. They are included in the SAFIRE-CLIMAT dataset. It is planned to estimate the sea surface temperature from these measurements.

### 3.2.3 Visible images (SAFIRE-CAMERA)

To visualize the context of the data acquired by in-situ measurements or remote sensing, two high-resolution cameras were mounted on the aircraft. One camera, an AV GT 1920C model with a resolution of 1936 $\times$ 1456 pixels and a wide angle (focal length of 4.8 mm), took high frequency images (10 frames per second) through the ATR window on the side of the horizontally-staring lidar and radar instruments. The other camera, a Mako G-223 model with a resolution of 2048 $\times$ 1088 pixels and a

focal length of 16 mm, looked down towards the sea surface at a moderate frequency (1 frame per second). The images taken through the aircraft windows often appear dark because the choice was made to avoid saturation due to the brightness of the clouds as much as possible, especially when the sun is behind the aircraft (Fig. 8a). The downward-looking camera can detect the presence of clouds below the aircraft and can help characterise the state of the ocean surface (Fig. 8b).

Three types of products are derived from these cameras: movies (in avi format) are produced for each camera ("window" or

"ground") and for each flight, and high-resolution images (in bmp format) are produced for the window camera for R and L patterns.

### 3.3 In-situ turbulence measurements (SAFIRE-TURB)

The 5-hole nose radome and specific temperature and humidity sensors mounted on the ATR (Rosemount and fine wire thermometers, Licor and KH20 hygrometers, see Tables 5 and 6 and section 3.1) measured rapid fluctuations of the three wind

components, temperature and humidity. Based on these high frequency (25 Hz) measurements, the SAFIRE-TURB turbulence dataset was produced to characterize the turbulent characteristics of the atmosphere through a number of diagnostics. The data processing strategies, the calibration methodologies, the procedures of quality control applied to the 25 Hz temperature and





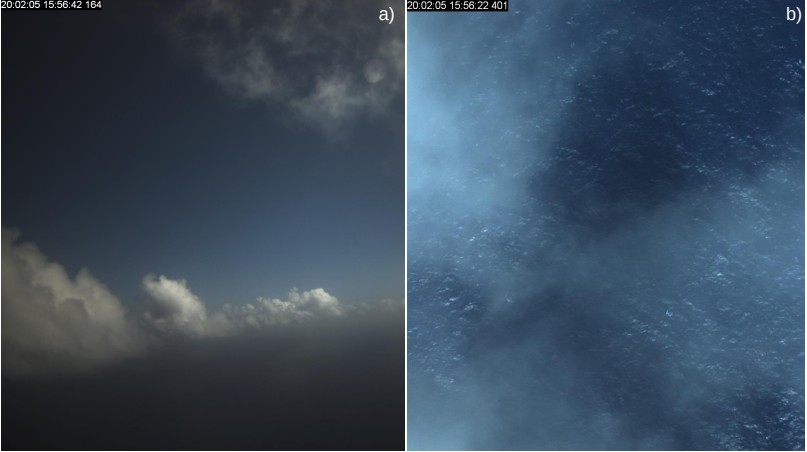

**Figure 8.** (a) Example of a cumulus scene captured by the visible camera through the aircraft window on 5 February 2020 at 15:56 UTC. (b) Image acquired a few minutes earlier by the camera looking down towards the ocean.

moisture measurements, and the methods used to estimate the turbulent diagnostics are explained in details in Brilouet et al. (2021).

The dataset includes two kinds of products: 'Turbulent fluctuations' and 'Turbulent moments'. The 'turbulent fluctuations' include time series of high-frequency fluctuations of the dynamical and thermodynamical variables over straight and stabilized segments of T-shortlegs, T-longlegs or T-longestlegs kind (Table 2). For each segment, the fluctuations time series are either detrended ('DET') or high-pass-filtered ('FIL') with a cutoff frequency of 0.018 Hz (about 5 km wavelength). The comparison of the 'DET' and 'FIL' calculations informs about the homogeneity of the sample and about random and systematic sampling

errors.

    The 'turbulent moments' include means, variances and covariances of dynamical and thermodynamical variables, turbulent kinetic energy and dissipation rate, third order moments and skewnesses of wind components, potential temperature and water vapor mixing ratio. They also include characteristic lengthscales such as the integral lengthscale or the wavelength of the vertical velocity density energy spectrum peak, error estimates on the turbulent moments, and quality flags on the temperature and

humidity measurements. These diagnostics are produced for each type of segment (T-shortlegs, T-longlegs and T-longestlegs).

    This dataset is produced for two levels of data processing. In the Level 2 dataset, the turbulent moments and fluctuations are calculated for each humidity sensor and each temperature sensor, and a quality flag is associated with each sensor. In the Level 3 dataset, a 'best estimate' of the turbulent moments and fluctuations is provided, together with a quality flag; for each segment, the best estimate corresponds to the moments and fluctuations computed from the sensor that has the best quality flag

over this segment. The dataset is distributed in NetCDF files whose nomenclature is summarized in Table 3 of Brilouet et al. (2021).



| Instrument | Brief description | Position on ATR | |
|---|---|---|---|
| UHSAS | optical-scattering aerosol particle spectrometer measuring aerosol sizes from 0.06 to 1 $\mu$m | pod under left wing; [CPCGB, FR18-19] | |
| FSSP300 | optical-scaterring aerosol and cloud particle spectrometer measuring particle sizes from 0.45 to 20 $\mu$m | fuselage corona (left hand side); [CPCGH, FR18-19] | |
| FCDP | optical-scattering cloud particle spectrometer measuring particle sizes from 1 to 50 $\mu$m | pod under right wing; [PDC] | |
| CDP-2 | optical-scattering droplet spectrometer measuring particle sizes from 2 to 50 $\mu$m | pod under left wing; [PGC] | |
| 2D-S | optical array stereo probe imager measuring particle sizes from 10 $\mu$m to 2 mm | pod under right wing; [PDC] | |
| LWC300 | hot wire probe measuring liquid water content up to 3 g m$-$3 | fuselage corona; [N1-5, FR13-14] | |

**Table 8.** Microphysical probes mounted on the ATR for EUREC[4]A. See Annex C for the correspondence between the position on the aircraft and the ATR configuration.



### 3.4 In-situ aerosol and cloud measurements

The ATR payload included a suite of six instruments to measure in-situ aerosol and cloud properties (Table 8). The Ultra High Sensitivity Aerosols Spectrometer (UHSAS), the Forward-Scattering Spectrometer Probe (FSSP300) and the LWC300 were operated by SAFIRE. The Cloud Droplet Probe (CDP-2), the Fast Cloud Droplet Probe (FCDP) and the 2D-Stereo (2D-S) are part of the Microphysics Airborne Plateforme (PMA), a French national facility operated by LaMP (Laboratoire de Météorologie Physique). Before take-off, all the data acquisition systems were synchronized to the aircraft central time database (GPS).

### 3.4.1 Aerosols (UHSAS)

A UHSAS-A probe (airborne version, serial no: 1303-007) was mounted on the lower left-hand pod on the fuselage section (Fig. 7). This probe is an optical-scattering aerosol particle spectrometer developed and commercialized by Droplet Measurement Technologies (DMT) that counts and sizes particles in the 0.06 to 1 $\mu$m range. The sizes are then sorted into 99 linearly spaced size bins of fixed width (9.7 nm).

The operating principle is as follows: the external air drawn at a controlled flow rate (about 50 sccm) enters the instrument optical detector, where it is aerodynamically focused and brought through a laser beam (Nd3+:YLiF4 laser operating at 1053 nm). The laser light scattered by each aerosol particle is collected by two pairs of Mangin collection optics and the scattered intensity is measured with a dual Avalanche photodiode/low-gain PIN photodiode detection system. The size of each particle is derived from the scattered intensity by using Rayleigh (40-300 nm) or Mie (300-1000 nm) scattering models implemented in the instrument (they are not corrected for variations in particle refractive index or non-sphericity). The UHSAS-A used in EUREC[4]A was last maintained and calibrated by DMT in December 2018 and a calibration check was performed at SAFIRE prior to the campaign in May 2019.

According to the manufacturer, UHSAS operation is limited to a non-condensing environnement. Ladino et al. (2017) reported that UHSAS measurements are subject to water contamination when performed in a cloudy area, which is also visible in our data. Therefore, UHSAS measurements made in cloudy area (determined by LWC $>$ 1 mg m$^{-3}$ using CDP and 2D-S data, as in the case of Ladino et al. (2017)) are rejected. Moreover, the UHSAS has a maximum count rate of 3000 per second and Cai et al. (2008) has shown that the detection efficiency decreases when the particle concentration exceeds 3000 cm$^{-3}$ due to coincidence effect. Therefore, points where the total count exceeds 3000 per second are removed from the data. According to Cai et al. (2008), particle concentrations in the small size range come with a caveat that the detection efficiency of a UHSAS (lab version) tends to decrease for particles smaller than 100 nm. Finally, inspection of the housekeeping data revealed erratic variations in the sample flow rate between 32 sccm and 50 sccm, caused by a loose electrical connection at a mass flow controller. Periods of large sample flow variation are manually identified and discarded. The aerosol concentration is calculated from the probe counts per second and the sample flow rate converted from mass (sccm) to volumetric flow rate (cm$^{-3}$) using temperature and pressures measurements from the aircraft core instruments (sections 3.1.2 and 3.1.3).

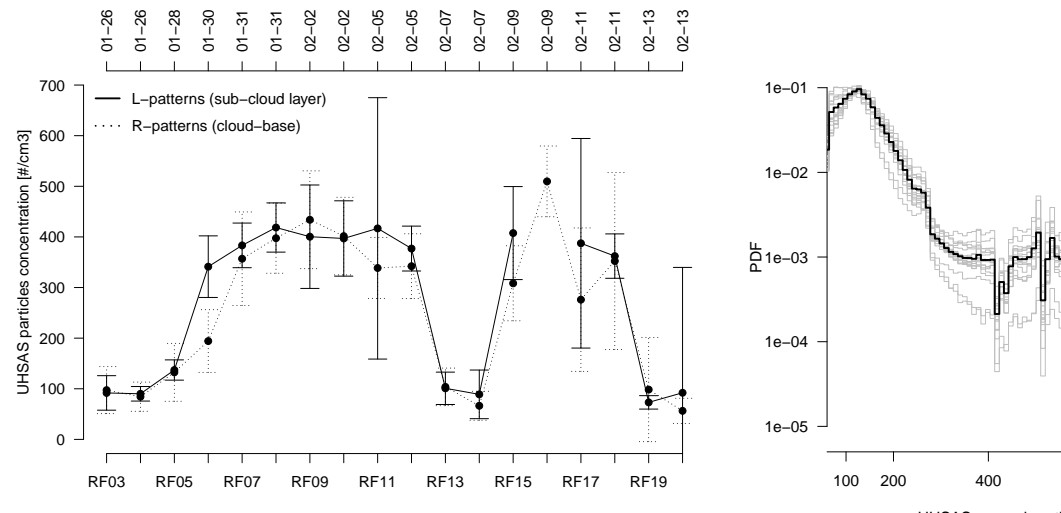

**Figure 9.** (left) Evolution of the total concentration of aerosol particles (per cm$^{-3}$) within the subcloud layer and at cloud base derived from UHSAS data (vertical bars represent the standard deviation across the different R-patterns or L-patterns during each flight). Note that the time axis is not linear and markers are only related by a line to ease readability. (right) Probability distribution function of the aerosol particle size derived from UHSAS data over the R-patterns flown at cloud-base; the mean of each ATR flight is shown in grey, and the mean over all flights in shown in black.

The total concentration of aerosol particles and the particle size distribution measured by UHSAS during the different ATR
flights is shown in Fig. 9. The concentrations in the subcloud layer and at cloud-base are generally similar, although a few flights (RF06, RF11, RF15 and RF17) show a slightly reduced concentration at cloud-base. In every case, the concentration is highly variable, with two main regimes: average aerosol concentrations are about 100 cm$^{-3}$ in half of the flights and about 300-400 cm$^{-3}$ in the other half. The particle size distribution also varies among flights, with the highest variability occurring in the frequency of large particles (diameters larger than 300 nm).

**3.4.2  Cloud microphysics**

Cloud microphysical measurements were made with two instruments: the CDP-2 which counts and sizes cloud droplets in the 2-50 $\mu$m size range, and the 2D-S which images cloud, drizzle and raindrops in the 10-1280 $\mu$m nominal size range (Table 8). Both instruments were mounted under the wings of the ATR, one on the right side and the other on the left side (Fig. 7). Throughout the campaign, the optics of the 2D-S and CDP-2 (and FCDP) probes were cleaned after each flight to remove
traces of dust and salt. At low altitudes where the air is warm, the temperature of the CDP-2 and 2D-S lasers increased rapidly and therefore the instruments were often switched off by the operator to avoid damaging the probe. As a result, few CDP-2 and





2D-S measurements are reported along the subcloud layer legs.

**CDP-2: cloud droplets**

The CDP-2 (serial no. 1711-111, equipped with anti-shatter tips) is a cloud particle spectrometer that counts and sizes cloud droplets in the 2-50 $\mu$m range and sorts them into 30 size categories with a resolution of 1-2 $\mu$m. The 1 Hz raw data (histograms of counts per second) are processed using DMT's built-in counting and sizing algorithms based on the Mie scattering model, assuming that droplets are spherical with a refractive index of 1.33, and converted to concentrations with the probe sample volume. The sample volume is calculated using the true air speed of the aircraft from SAFIRE-CORE data and the calibrated

sample area ($0.292 \text{m}^2$) determined prior to the campaign by mapping the probe's response to calibrated water microdroplets injected across the laser beam with an apparatus similar to Lance et al. (2010). At 100 m s$^{-1}$, which was the typical ATR airspeed during the scientific flights, the sample volume was about 30 cm$^3$ s$^{-1}$. The calibration of the CDP-2 with respect to particle size was regularly monitored during the campaign by means of calibrated glass bead injection tests.

    Measurements in the subcloud layer reveal that the CDP-2 can detect non-cloud droplet particles such as large/ultra-large

aerosols. Although these particles may not satisfy the underlying assumption of the CDP-2 sizing algorithms, it was decided not to filter out these measurements in the CDP-2 files so that further investigations of large aerosols may be conducted, at least qualitatively. However, the response of the CDP-2 to such aerosol particles being unknown, the data taken in non-cloudy areas are subject to unquantified errors.

**2D-S: cloud droplets, drizzle and raindrops**

    The 2D-S (serial no: 006) is an optical array probe imaging cloud, drizzle and rain particles in the range 10–1280 $\mu$m (the stereo capability of the probe is not used here): an array of 128 photodiodes is illuminated by a laser sheet; when a hydrometeor crosses the sample area (about 0.128 cm $\times$ 6.3 cm, located between a pair of emitting/receiving arms), it shades some of the photodiodes. The binary state (occluded/non occluded) of the photodiodes is recorded at high frequency (up to 17 MHz for

this probe), producing time-discretized black-and-white slices of the particle's silhouette which are subsequently concatenated to reconstruct a projected 2D black-and-white image of the hydrometeor with a resolution of 10 $\mu$m.

    The raw data (from either vertical or horizontal channel, whichever worked best during the flight) is processed using the LaMP in-house processing routines which stem from the early release of the SPEC 2DSView software and are continually updated to integrate state-of-the-art corrections.

The calculation of the sample volume takes into account the decrease in field depth with particle size and follows the manufacturer's formula given in Lawson et al. (2006) and the overload periods of the probe. Artifacts due to noisy or dead pixels are identified and removed using the pixel analysis described in Lawson (2011). This probe is equipped with anti-shattering arm tips (K-tip, Korolev et al. (2013)) designed to prevent ice/droplet fragments from falling into the probe sample volume and contaminating the measurement at the lower end of the size spectra (note that no ice was sampled along the ATR

flights of EUREC$^4$A). In addition to the K-tip, a splash/shatter detection and removal algorithm based on arrival time analysis is applied (e.g. Field et al. (2006), Korolev and Field (2015)). The size of particles seen out of focus is corrected using the





Korolev (2007) diffraction correction. Despite these efforts to clean artifacts, the concentration in the first few bins remains questionable for reasons described in Thornberry et al. (2017) and Bansemer (2018) (the contribution of remnant noisy events is amplified in the concentration calculation due to the small sample volume). The size of truncated particles (partial images)

is corrected according to Korolev and Sussman (2000) and the nominal size range (10-1280 $\mu$m) is extended to 2.56 mm in post-processing.

Once most of the artifacts have been corrected, a series of geometrical descriptors, e.g. size (defined here as the diameter of a circle having an area equal to the projected area of the particle, often referred to as surface equivalent diameter in the literature, $D_{eq}$), area or perimeter are retrieved from each individual 2D image. Statistical properties are then calculated at 1 Hz, such

as the particle size distribution (PSD) or the total concentration (calculated as the sum of bin concentrations). The mass size distribution (MSD) is computed from the PSD assuming that the particles are spherical with a liquid water density of $1\,\mathrm{g\,cm^{-3}}$.

**PMA composites**

As the cloud drop size distribution is broad, a combined PMA dataset is produced that merges the CDP-2 and 2D-S data

into a single composite spectrum that ranges from 2 $\mu$m to 2.55 mm, at the native size resolution of the CDP-2 up to 43 $\mu$m, the 10 $\mu$m size resolution of the 2D-S up to 1 mm and a coarser resolution of 100 $\mu$m from 1.05 up to 2.55 mm. We define a cloud mask and a drizzle mask based on the liquid water content (LWC) and the particle size (diameter D): a cloud particle is identified when the LWC of droplets smaller than D0 exceeds LWC0, where LWC0 and D0 are specified thresholds of LWC and D, respectively. There is no simple definition of cloud situations, and therefore the values of these thresholds remain

uncertain. Here, we use LWC0 = $0.010\,\mathrm{g\,m^{-3}}$ (which is consistent with other observational and modeling studies of trade-wind clouds such as Heymsfield and McFarquhar (2001) or vanZanten et al. (2011)) and D0 = 100 $\mu$m (which is consistent with the AMS glossary definition of cloud drops as water particles between 1 and 100 $\mu$m in diameter). We assume that drizzle occurs (drizzle mask is set to 1) when $100 \leq D < 500$ $\mu$m, and rain occurs when $D \geq 500$ $\mu$m.

The cloud LWC was inferred from the size distribution of cloud particles measured by the CDP-2 and 2D-S probes. It

was also measured independently by a hot wire probe (DMT LWC-300) that was part of the core instrumentation of the ATR (Table 8, note that the LWC300 sensor broke during RF14 and was immediately replaced by a new one). The hot wire estimates the LWC by measuring the heat released by the vaporization of water droplets on a heated cylinder exposed to the airstream. This calculation is made with the Particle Analysis and Display System (PADS) software, using the aircraft airspeed, pressure and deiced temperature measured by the ATR and the formulas given in the DMT PADS Manual Hot Wire Module

3.5.0 DOC-0290 Rev A. However, the collection efficiency of the sensor is limited for small droplets ($< 10$ $\mu$m) and the evaporation of large drops ($> 50$ $\mu$m) can be incomplete, which can underestimate the LWC measurement in drizzle and rain conditions (DMT LWC-300 LWC operator's manual DOC-0361 Rev C). The LWC estimate derived from the CDP-2 and 2D-S probes (distributed in the PMA composite dataset) is thus considered to be more precise than that derived from the LWC-300 (distributed in the SAFIRE-CORE dataset).

Cloud droplet number concentrations at cloud base, and their relationships with aerosol number concentrations (derived from UHSAS) are shown on Fig. 10a. Cloud droplet number concentrations tend to be about 2/3 of the aerosol concentrations, with a

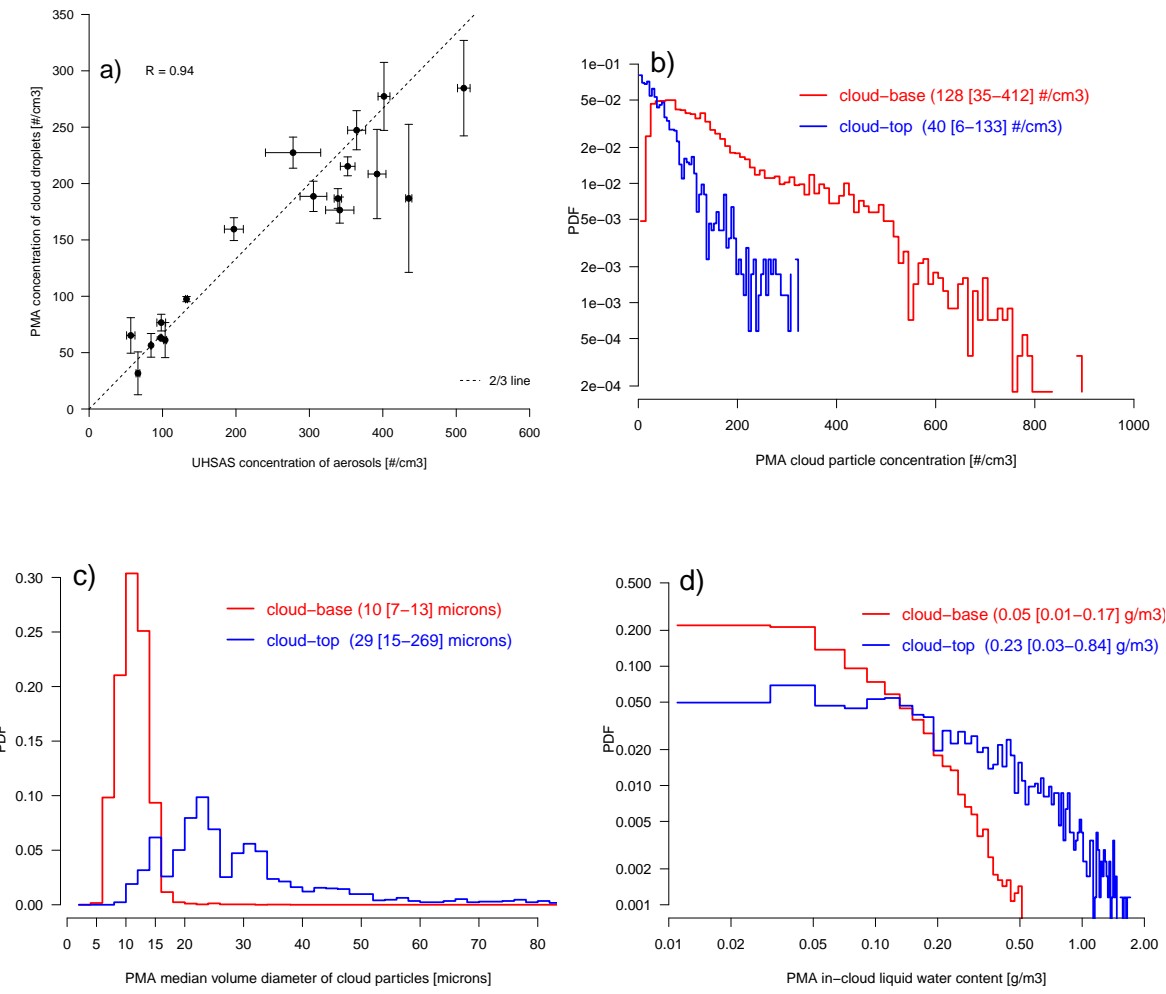

**Figure 10.** (a) Relationship between the concentration of aerosols (from the UHSAS dataset) and the concentration of cloud droplets (from the PMA dataset, excluding drizzle and rain particles) calculated for the R-patterns flown at cloud-base during the whole EUREC[4]A campaign. The mean of each ATR flight is reported together with the standard deviation among the different R-patterns of the flight. Other panels (b-c-d) show the probability distribution function (calculated over all the R-patterns flown at cloud-base or at cloud-top) of (b) the total concentration of cloud particles (c) the median volume diameter (MVD) of cloud particles, and (d) the in-cloud liquid water content (LWC) derived from the composite PMA dataset. The mean of each quantity is reported, together with the $10^{th}$ and $90^{th}$ percentiles of each distribution (in brackets). (b-d) histograms are calculated for in-cloud conditions (where cloud particles can coexist with drizzle or rain particles).





strong case-to-case co-variability (correlation of 0.94). At larger aerosol concentrations, the cloud droplet concentrations tend to disproportionately decrease. This could be indicative of a lower maximum of cloud-base supersaturations in an aerosol rich environment, or a less cloud-active aerosol in conditions when the concentrations are high. The distribution along all the R-

patterns of the droplet number concentration, MVD and LWC values of the clouds derived from the PMA composite dataset is shown on Fig. 10b-d. Cloud particle concentrations are very variable but, on average, they tend to be much larger at cloud base (median of 128 cm$^{-3}$) than in the stratiform layers of trade-cumuli detraining near the inversion level (median of 46 cm$^{-3}$); on the other hand, cloud particle sizes and cloud liquid water contents tend to be much smaller at cloud base (about 10 $\mu$m and 50 mg m$^{-3}$, respectively) than at cloud top (about 24 $\mu$m and 200 mg m$^{-3}$ near the inversion level). The range of MVD values

measured near cloud base and cloud top during EUREC[4]A are similar to those measured in trade cumuli over the Indian ocean (Heymsfield and McFarquhar, 2001) or in cumulus clouds over the sea around the United Kingdom (Raga and Jonas, 1993).

### 3.4.3 Datasets

An aerosol dataset was produced on the basis of UHSAS measurements. It is distributed as an ensemble of NetCDF files (one file per flight) that include products such as the Particle Size Distribution (PSD) and the total concentration of particles (NT),

all processed at a frequency of 1 Hz.

A cloud dataset was produced on the basis of CDP-2 and 2D-S measurements (future versions of the dataset might include data from the FSSP-300 and FCDP probes). It is distributed as a set of NetCDF files (one file per flight) which include the following products: particle size distribution (PSD), total particle concentration (NT) and liquid water content (LWC, assuming particles are spherical with a density of 1 g cm$^{-3}$), all processed at a frequency of 1 Hz.

The data are distributed for two levels of processing: the level 2 dataset is associated with single instruments (either 2D-S or CDP-2) while the level 3 dataset corresponds to a combined PMA dataset that merges CDP-2 and 2D-S data into a single composite spectrum that spans the range 2 $\mu$m to 2.55 mm. The composite dataset includes additional products such as a cloud mask, a drizzle mask and a rain mask (defined in section 4.3), as well as the 6th moment of the particle size distribution to ease the comparison with radar reflectivities. The periods of flight when the probes are switched off are filled with NaN values. All

datasets also include the time and aircraft position from the SAFIRE-CORE dataset.

The LWC measurements from the LWC-300 are included in the SAFIRE-CORE dataset at 1 Hz.

### 3.5 Lidar and radar remote sensing

### 3.5.1 Horizontal lidar measurements (ALIAS)

To characterize the presence of clouds and aerosols in the lower troposphere, the ATR was equipped with a lightweight

backscatter lidar named ALiAS (Airborne Lidar for Atmospheric Studies) emitting at the wavelength of 355 nm and detecting polarization (Table 9, Chazette et al. (2012, 2020)). The main role of this lidar was to measure, together with the BASTA radar, the fractional area covered by the cloud field near the cloud-base level. For this purpose, the line of sight of the lidar was oriented horizontally, looking through one of the ATR windows (UV fused silica glass) on the right side of the aircraft (Fig. 7).





| Instrument | Brief description | Position on ATR | |
|---|---|---|---|
| ALiAS | Horizontally-staring backscatter lidar operating at 355 nm and detecting polarization | window (right-hand side); [H36, FR34-35] | |
| BASTA | Horizontally-staring bistatic FMCW 95 GHz Doppler cloud radar | window (right-hand side); [H28-H30, FR30-32] | |
| RASTA | Two antennas zenith and backward-looking 95 GHz Doppler pulsed cloud radar | [FAr-OH4, FR32-33] | |
| Picarro | L2130-i cavity ring-down laser spectrometer | window (right-hand-side); [H6, FR19-20] | |

**Table 9.** Lidar-radar remote sensing and stable isotopologue measurements. See Annex C for the correspondance between the position on the aircraft and the ATR configuration.

The native resolution of the lidar backscatter profile along the line-of-sight is 0.75 m. However, to improve the signal to noise ratio, a low-pass filter has been applied and the resolution was downgraded to 15 m. In addition, the backscatter profile was averaged over 50 consecutive shots during the acquisition, which corresponds to approximately one recording every 5 s





(averaging time 2.5 s and recording time 2.5 s). The backscatter lidar observations are used to define a cloud mask in the direction perpendicular to the aircraft trajectory. In this direction, the signal was distinguishable from noise up to a distance of about 8 km in clear-sky conditions. However, this range was reduced in the presence of strong scattering, for instance from

thick clouds. It means that during the R-patterns, as the aircraft was flying rectangles of about 120 km (along track) × 20 km (cross track), the lidar was able to sample most of the rectangle area unless thick clouds within the rectangle extinguished the lidar signal at some distance of the aircraft.

Both aerosol and cloud products have been derived from the ALiAS observations, and the data are distributed as a set of NetCDF files (one per flight) for different levels of processing. Level 1 provides the raw profiles at native resolution recorded

by the acquisition system. Level 1.5 data are geolocated, calibrated and corrected for geometric factors and molecular transmission, and time series of the apparent backscatter coefficient (ABC) and Volume Depolarization Ratio (VDR) are produced with a resolution of 15 m along the lidar line of sight. Level 2 provides cloud and aerosol detection information and products, including a cloud mask and an aerosol extinction coefficient (AEC) along the horizontal line of sight. Level 3 provides statistics about the length of the cloud chords inferred from the lidar cloud detection. The ALiAS dataset is described in detail in

Chazette et al. (2020).

Fig. 11 shows the relationship between the total concentration of particles measured by the UHSAS microphysical probe (section 3.4.1) and the aerosol extinction coefficient retrieved from ALiAS lidar data. Although the two instruments sample the atmosphere very differently (the lidar probes the atmosphere horizontally perpendicular to the aircraft trajectory over a range of several kilometers while the UHSAS probes the atmosphere in-situ along the aircraft trajectory), the two measurements

are highly correlated (R = 0.89). It shows the consistency of the two measurements and confirms the strong variability of the atmospheric load in aerosols during the campaign (Fig. 9).

### 3.5.2   Horizontal radar measurements (BASTA)

To characterize the cloudiness in synergy with the lidar, an horizontally-staring cloud radar named BASTA (Bistatic Radar System for Atmospheric Studies) was mounted on the right-hand side of the ATR (Table 9). BASTA is a 1 W bistatic FMCW

(Frequency Modulated Continuous Wave) 95 GHz Doppler cloud radar developed from the ground-based BASTA system (Delanoë et al., 2016). It was used in an aircraft for the first time during EUREC⁴A, with two antennas of 20 cm (0.95° beamwidth) installed in back lateral windows of the ATR (Table 9). The radar was operated in two modes, one after the other, at 12.5 m and 25 m range resolutions with 0.5 and 1 s time resolutions respectively. It led to a measurement in one mode every 1.5 s. The maximum range was 12 km with an ambiguous velocity of 9.85 m.s⁻¹ for both modes. The minimum detection

range is about 80 m from the aircraft due to coupling between the two antennas.

The Level 1 of BASTA product contains, for both modes, the calibrated and range corrected radar reflectivity, the Doppler velocity and a mask distinguishing the meteorological target from background noise and surface echoes. The calibration of the radar has been derived from other field campaigns and confirmed using in-situ data (a reflectivity was calculated from the CDP and 2D-S cloud particles data and compared with radar measurements in cloudy conditions). The sensitivity of the

radar is estimated at around -35 dBZ at 1 km. Level 2 data are the most elaborated product, the two modes being combined

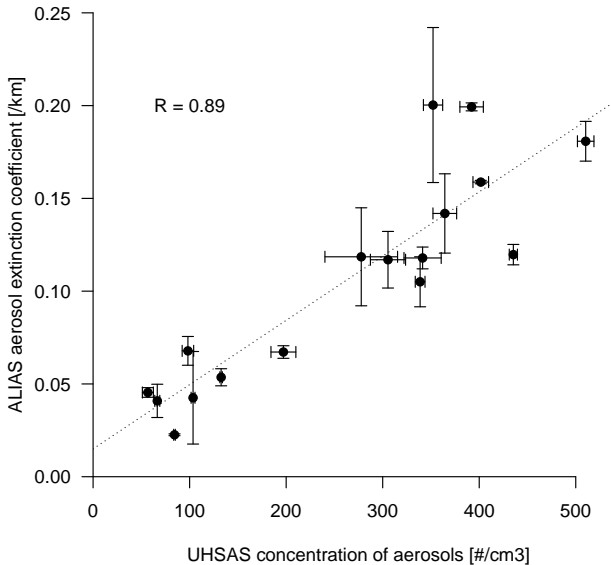

**Figure 11.** Relationship between the total concentration of aerosols measured by UHSAS at cloud-base and the aerosol extinction coefficient measured by the horizontally-pointing ALiAS lidar at cloud base. Horizontal and vertical bars represent ± the standard deviation of measurements across the different R-patterns of each flight.

to optimise the advantages of each range resolution. Within the first 250 meters, the 12.5 m mode is used while the 25 m mode covers the rest of the profile. The combined reflectivity and Doppler profiles are available every 1.5 s. The radar gates are geolocated in latitude, longitude and altitude in order to derive maps. The reflectivity is corrected for gaseous attenuation using colocated information from dropsonde temperature, humidity and pressure. A parameterisation of liquid attenuation for

both cloud and precipitation as a function of reflectivity was derived thanks to in-situ data and applied to correct reflectivity for liquid attenuation. The corrected reflectivity is then used to distinguish cloud areas from drizzle or rain (section 3.5.3). The radar Doppler velocity is corrected for aircraft motion and folding using gate-to-gate correction. All files are available in a self-documented NetCDF file.

### 3.5.3  Combined lidar-radar measurements (BASTALIAS)

Based on ALiAS and BASTA data, a combined dataset was developed that takes advantage of the lidar-radar synergy and complementarity to improve the detection of clouds, drizzle and rain (Fig. 12).

For this purpose, the two modes of the BASTA radar products are merged on a single horizontal grid (resolution of 12.5 m within the first 200 m from the aircraft, and 25 m beyond this distance), and a single time resolution (1.5 s). Then the reflectivity is corrected for liquid and gas attenuation and the radar sensitivity is defined as a function of the distance from the aircraft.





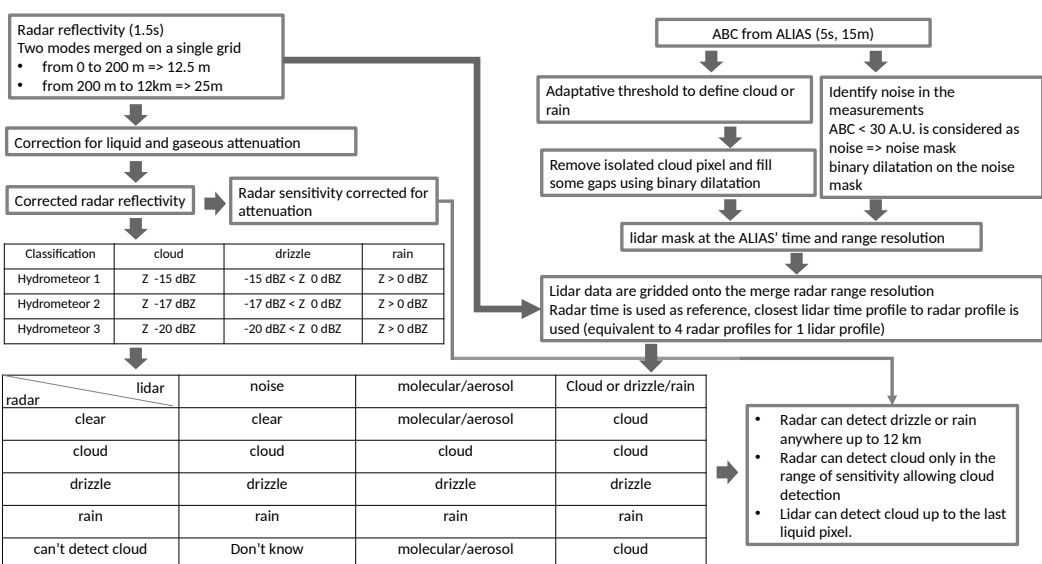

**Figure 12.** Cloud detection algorithm applied to BASTA+ALiAS data to detect hydrometeors (clouds, drizzle and rain).

A first classification of hydrometeors is then made on the basis of radar observations. As the reflectivity associated with the presence of a remote hydrometeor depends on the drop diameter, reflectivity thresholds can be used to distinguish cloud droplets from drizzle or rain. The definition of these thresholds differs across ground-based radar studies; the threshold distinguishing clouds from drizzle ($Z_d$) is often set at -20 dBZ (e.g. Kato et al. (2001)), but it can also be set at -17 dBZ or -15 dBZ. The BASTALIAS dataset thus considers three options for the definition of cloud droplets, associated with each of these thresholds.

The threshold distinguishing drizzle from rain is set at $Z_r$ = 0 dBZ.

     To assess the ability of these reflectivity thresholds to distinguish between cloud, drizzle and rain situations, we calculate the reflectivity $Z_{\mathrm{PMA}}$ that would correspond to the drop size distribution of the PMA dataset. It is done using the T-matrix approach and accounting for the beam orientation and for the non-sphericity of large particles (for the smallest particles, $Z_{\mathrm{PMA}}$ follows Rayleigh theory and is equal to $10 log_{10}(M6)$, where M6 is the $6^{th}$ moment of the drop size distribution). The

distribution of $Z_{\mathrm{PMA}}$ values for situations classified by the PMA microphysical masks (based on LWC and D measurements, section 3.4.2) as cloud-only, drizzle-only or rain-only shows that clouds and rain are mainly associated with reflectivities lower than -20 dBZ and larger than 0 dBZ, respectively (Fig 13), which supports the $Z_d$ and $Z_r$ thresholds used in the BASTALIAS dataset. Reflectivities between -20 and 0 dBZ are predominently associated with drizzle. However, drizzle is associated with a broader range of reflectivities and therefore its identification from reflectivity thresholds remains imperfect.

Since the sensitivity of the radar decreases as the distance from the aircraft increases, BASTA can only detect clouds within a limited distance from the aircraft; beyond this point, the radar can only detect drizzle or rain. The range over which the radar can possibly detect clouds ($D_{\mathrm{cloud}}^{radar}$) is determined by the distance at which the expected radar sensitivity corrected for attenuation



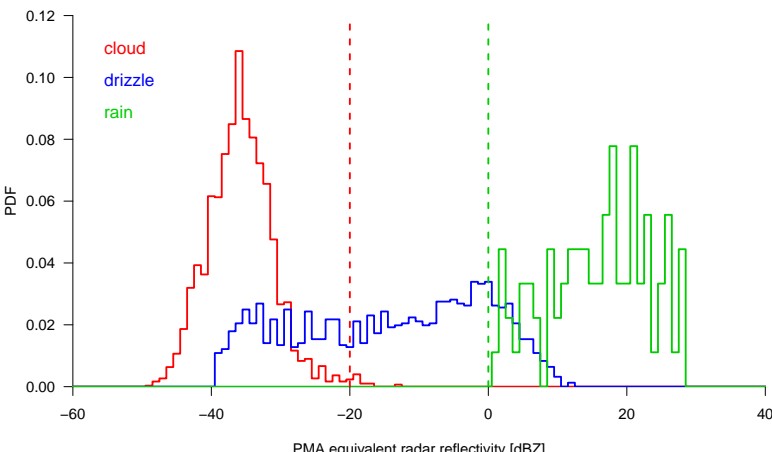

**Figure 13.** Probability distribution function of equivalent radar reflectivities calculated for each ATR flight from the PMA particle size distribution for situations defined as cloud-only, drizzle-only or rain-only by PMA masks (based on LWC and cloud drop diameter, section 3.4.2). The PMA measurements were performed along the R-patterns flown around the cloud-base level.

equals $Z_d$. In the trade-wind boundary layer conditions of EUREC$^4$A, the radar could detect clouds over a maximum horizontal distance ranging from 1.5 to 3.5 km (2.2 km on average) for $Z_d$ = -20 dBZ and from 2.9 to 6.2 km (3.8 km on average) for $Z_d$
= -15 dBZ. On the other hand, drizzle and rain could be detected at any distance up to 12 km if there is no rain in the vicinity of the aircraft.

In parallel, the ALiAS lidar data at their original resolution (level 1.5 data from Chazette et al. (2020)) are analyzed to determine the horizontal lidar profile that corresponds to the molecular or aerosol backscatter, to estimate the noise level, and to detect the presence of clouds. The lidar cloud detection methodology used in the BASTALIAS dataset is inspired from that
developed for the Calipso space lidar and the airborne LNG lidar (Ceccaldi et al., 2013). Although derived from a different methodology, the lidar-only cloud mask of the BASTALIAS dataset is very consistent with that proposed by Chazette et al. (2020), showing the robustness of the cloud detection from lidar measurements. This information is then used to define a lidar pseudo cloud mask at the same space and time resolution as the radar information (for this purpose, each radar time is associated with the closest lidar observation in time). The cloud detection by the lidar is considered impossible beyond the
distance from the aircraft ($D_{\mathrm{cloud}}^{\mathrm{lidar}}$) at which the lidar backscatter signal is completely extinguished or undistinguishable from noise. During EUREC$^4$A, $D_{\mathrm{cloud}}^{\mathrm{lidar}}$ ranged from 0 to 8 km, and was about 5 km on average.

Finally, the lidar and radar cloud masks are analyzed jointly to make a final classification of hydrometeors and a lidar-radar cloud mask (Fig. 12). The synergy between lidar and radar is illustrated by two examples of individual radar and lidar profiles along their horizontal line of sight (Fig. 14). In the first one, derived from a flight (RF05) associated with small very shallow
clouds ('Sugar', Table 3), the lidar detects three clouds in a row which are not detected by the radar; beyond $D_{\mathrm{cloud}}^{\mathrm{lidar}}$ (6.3 km),



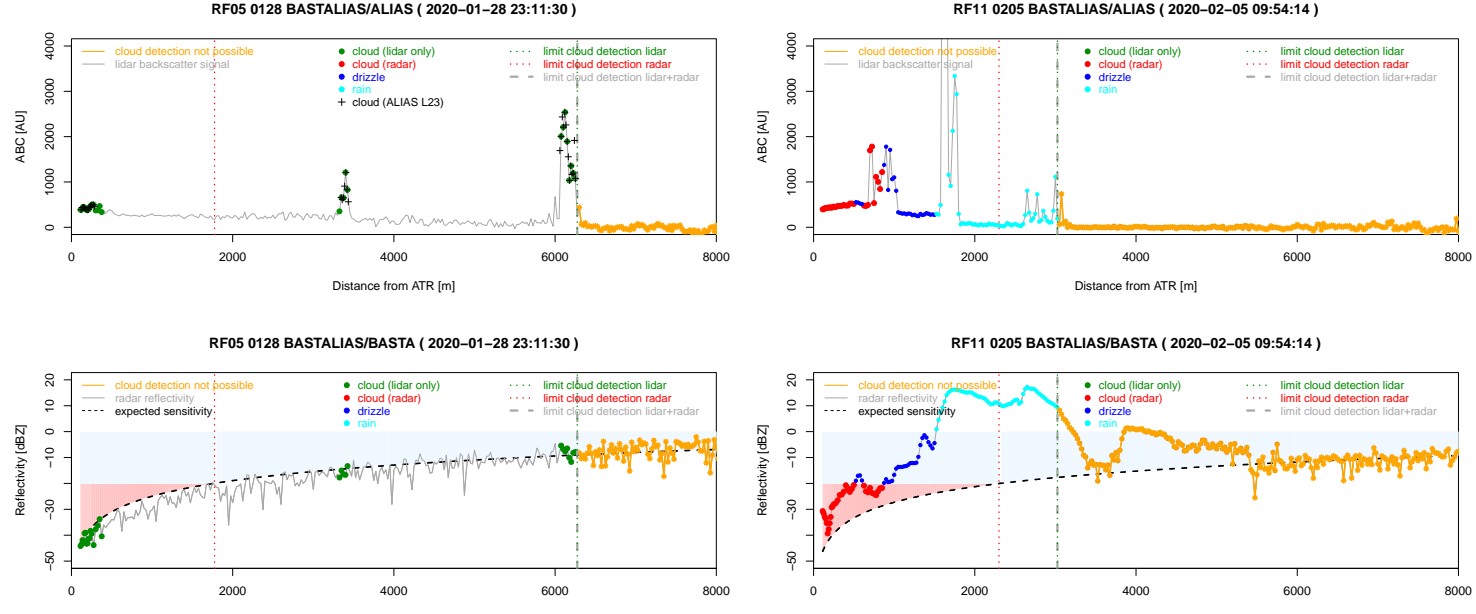

**Figure 14.** Illustration of cloud, drizzle and rain detection by horizontal remote sensing using lidar-radar synergy. The maximum distances $D_{cloud}^{lidar}$ and $D_{cloud}^{radar}$ over which cloud detection is possible with the ALiAS lidar or BASTA radar are indicated by green dash-dot and red dotted vertical lines, respectively. The range over which hydrometeor detection is no longer possible with radar or lidar is indicated in orange –it corresponds to the maximum of ($D_{cloud}^{lidar}$, $D_{cloud}^{radar}$). The classification of hydrometeors is reported on the lidar and radar signals: drizzle, rain, clouds detected only by lidar, and clouds detected by radar or both radar and lidar. On Jan 28th, 2020 (RF05) at 23:11:30 UTC, the lidar (ALiAS, upper left panel) detects three areas of strong backscatter along its line of sight, while the radar (BASTA, lower left panel) detects no hydrometeor in the range (0-1.8 km) in which it could possibly detect clouds; the areas of strong lidar backscatter therefore correspond to the presence of thin clouds; beyond 6.3 km, the lidar signal is fully extinguished and cloud detection is no longer possible. The cloud detection from ALiAS Level 2 and Level 3 (L23) dataset (performed at a horizontal resolution of 15 m, as opposed to 25 m for BASTALIAS) and using the methodology described in Chazette et al. (2020)) is also reported (note that the ALiAS L23 times have to be shifted by $-10$ s to coincide with those of BASTALIAS). On Feb 5th, 2020 (RF11) at 09:54:15 UTC, the lidar (upper right panel) measures four areas of strong backscatter and is fully attenuated beyond 3 km. The radar reflectivity (bottom right panel) shows that the first area corresponds to the presence of clouds, but that the following areas correspond to the presence of drizzle or rain (the cloud-drizzle and drizzle-rain transitions are defined by reflectivity thresholds, set here at -20 dBZ and 0 dBZ, respectively). In this case, no cloud can be detected beyond 3 km. This case is from an R-pattern flown near the inversion level; no cloud mask is available from the ALiAS L23 dataset on R-patterns above cloud-base.

the backscatter signal is extinguished and the cloud detection becomes impossible. In the second example, from RF11, the radar detects a cloud within the first kilometer, and then drizzle and rain. Wherever the radar detects drizzle or rain, the hydrometeors detected by the lidar are not considered as a cloud in the cloud mask.



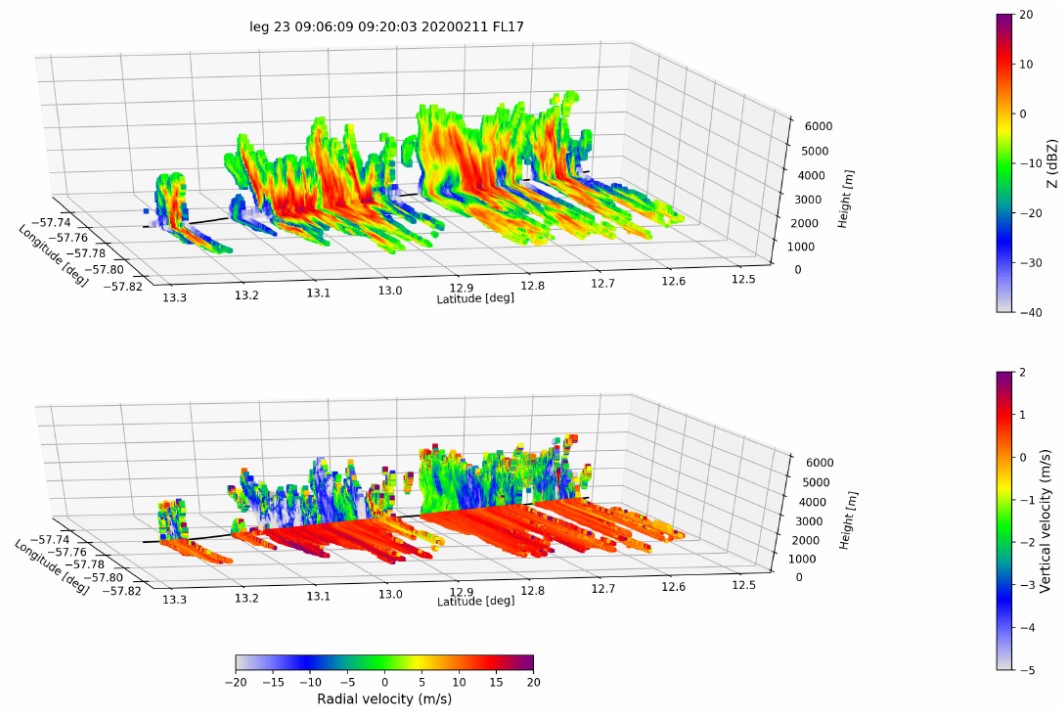

**Figure 15.** (top) Reflectivity corrected for attenuation (in dBZ) and (bottom) Doppler velocity (vertical component $V_z$ and radial component $V_x$, in m s$^{-1}$) from the vertically-pointing RASTA radar and from the horizontally-pointing BASTA radar displayed as a function of height (for RASTA) or horizontal range (for BASTA), along a subcloud-layer leg during RF17 (from 09:06:09 to 09:20:03 UTC on Feb 11 2020).

### 3.5.4 Vertical radar measurements (RASTA)

RASTA is an up-looking pulsed 95 GHz Doppler cloud radar with two antennas (zenith and up backward –with an elevation of 66.7°–, 30 cm large, Table 9). The radar was dedicated to the characterization of cloud microphysics and dynamics. The radar was operated at 30 m resolution with a maximum range of 6 km at 1 s integration. Both Doppler moments (reflectivity and velocity) and spectrum are available. As for BASTA, the radar reflectivity is range-corrected and calibrated, and the background noise is removed using a thresholding technique based on the background noise characteristics. The derived mask

is refined thanks to some image processing. The reflectivity is corrected for gaseous attenuation using colocated information from dropsonde temperature, humidity and pressure. Once the Doppler velocity is unfolded and corrected from aircraft's motion and when backward and zenith antennas are simultaneously available, the vertical velocity and the along-track wind components of the cloud/precipitation wind are retrieved. Two antennas allow us to retrieve the two components of the wind in the plane defined by the two antennas.

Level 2 data are distributed as a set of NetCDF files for the flights during which the radar was operating and clouds were detectable with the radar (RF03, RF04, RF11, RF12, plus all flights from RF13 to RF19). For all these flights but two (RF11





and RF12), two antennas were working (zenith and up-backward), which allows us to derive wind information (its radial component) in addition to cloud information. For RF11 and RF12, only one antenna (zenith) was working and therefore the wind information is not available.

Fig. 15 shows the reflectivity and Doppler velocity measured in the vertical and radial directions by the RASTA and BASTA radars during a leg of RF17. During this flight, the height of precipitating cloud tops could exceed 3 km. The vertical and radial reflectivity structures tend to reflect each other, suggesting a well-defined geometry (or aspect ratio) of the clouds. The vertical structure of the Doppler velocity from RASTA exhibits a maximum positive velocity near cloud top and negative velocities in the parts of clouds that are associated with falling hydrometeors (rain or drizzle).

### 3.6   Water stable isotopes (Picarro)

In addition to characterizing the meteorological, turbulent, microphysical, cloud and radiative properties of the atmosphere, the ATR measured the water isotopic composition of the atmosphere using a customised fast response cavity ring-down spectrometer from Picarro (version L2130-i). This effort took place as part of a wider EUREC[4]A-iso initiative involving multiple platforms and instruments (Stevens et al., 2021; Bailey et al., submitted). The rationale for isotopic measurements is that by

quantifying the relative content of isotopically heavy ($^1H^2H^{16}O$, $^1H_2^{18}O$) and light ($^1H_2^{16}O$) water molecules in the atmosphere, it is possible to get information about the transport, mixing and phase changes of water. Isotopically heavy water molecules are associated with lower saturation vapor pressures and smaller diffusion velocities than their most abundant, lighter counterparts. Therefore, the three main components of the boundary layer moisture budget, namely ocean evaporation, convective drying and moistening by hydrometeor evaporation, carry a distinct stable water isotope signature (Risi et al., 2019). Specificities in the

water vapor cycling associated with different mesoscale cloud organisation patterns, therefore result in characteristic isotopic fingerprints (Aemisegger et al., 2021b). Whether these fingerprints are primarily due to the local processing of water vapor in the marine boundary layer, or result from the interaction with the large-scale flow, is one of the questions to be addressed with water isotope tracers.

The isotopic composition of atmospheric water vapor on board the ATR was measured with a sampling frequency of 1 Hz

(Table 9). The CRDS system uses laser absorption spectroscopy as a working principle: the different isotopic molecules having different rotational-vibrational energy level structure, they exhibit different transition frequencies in the near-infrared region of the spectrum. Three nearby absorption peaks in the near-infrared region (7199–7200 $cm^{-1}$) corresponding to the three molecules ($^2H^1H^{16}O$, $^1H_2^{18}O$ and $^1H_2^{16}O$) are thus scanned by a laser in continuous wave operation mode. Laser light is injected through a semi-transparent mirror into a 35 $cm^3$ cavity with three mirrors in ring configuration (Crosson, 2008). A

photodetector is placed behind another mirror and measures the light intensity leaking out of the cavity. The isotope concentration is determined by measuring the exponential ring-down time of the laser intensity after the laser source has been switched off. The higher the heavy isotope concentration, the faster the decay of the laser intensity.

In the ATR, a rearward facing 30 cm long stainless-steel inlet with $\frac{1}{4}$-inch outer diameter was fitted to one of the front windows on the right-hand side of the aircraft (Fig. 7, Table 6). A 1.5 m heated PTFE line with 10 mm inner diameter was

flushed with an inlet pump (KNF, HN022AN.18) at a rate of 13 $Lmin^{-1}$. A filter (0.2 $\mu$m PTFE vent filter) was installed at





the end of the inlet line to prevent particles from entering the laser spectrometer. A subsample from the inlet line was drawn into the instrument by a second pump (KNF, N920AP.29.18). The flow rate through the laser system was 280 mL min$^{-1}$ and the residence time of the vapor sample in the system was 7-14 s, depending on the ambient pressure. This bottom-up residence time estimate from the gas flow setup was confirmed by a correlation analysis of the water vapor mixing ratio measured by the

Picarro and the ATR's dew point hygrometer. A time lag of between 6 s and 15 s was found for the Picarro with synchronized computer clocks and was corrected in the post-processing. More details on the setup can be found in Bailey et al. (submitted).

To assess the instrument's precision and drift, calibration gases were measured on the ground pre- and post-flight using a Picarro Standards Delivery Module (SDM). The high precision liquid pumps of the SDM deliver a thin stream of liquid water of known isotopic composition into a vaporiser heated to 140 °C. In the vaporiser, the liquid water droplets are completely

evaporated in a dry air stream, which was produced by pumping ambient air through a drying unit (Drierite) using a small air pump. In addition to the ground-based calibration runs, four in-flight calibrations were performed to assess the impact of aircraft vibrations on the precision of the measurements.

Recent studies have indicated that the precision of laser spectrometers in laboratory settings is comparable to the one of conventional isotope ratio mass spectrometer systems. However, for atmospheric field applications, the overall measurement

uncertainty can result from a range of factors such as calibration, sensitivity to variations in water concentration, and retention effects from the tubing (Aemisegger et al., 2012). A detailed post-processing procedure was therefore applied to account for these factors. In particular, a two-stage correction procedure following Weng et al. (2020) was applied at water vapor mixing ratios lower than 10'000 ppmv to correct for a known concentration-bias in laser spectrometric isotope measurements. The water vapor mixing ratio measurement from the CRDS system was calibrated based on a linear correction determined in the

laboratory using a dew point generator. More details on the post-processing are available in Bailey et al. (submitted). The dataset is distributed on AERIS as an ensemble of self-documented NetCDF files (Aemisegger et al., 2021a).

## 4   Consistency among observations

The ATR measured humidity, winds and clouds with multiple instruments based on different observation techniques. This redundancy and/or complementarity is an opportunity in several respects. It is an asset for the quality control of the data from

each instrument and for the processing of combined datasets taking advantage of the complementarity of the instruments. It also allows the robustness and the statistical representativeness of the measurements to be assessed. This last point is particularly important for EUREC[4]A, as the experiment was designed on the premise that the relationships between clouds and their environment could be characterized by combining measurements from several instruments and/or observing platforms that sample the atmosphere differently (Bony et al., 2017; Stevens et al., 2021).

The objective of this section is to verify this premise by comparing some of the main ATR measurements made by different instruments using different techniques and/or samplings. We also assess the consistency between the ATR measurements and the simultaneous dropsonde measurements (George et al., 2021) or BCO ground-based observations (Stevens et al., 2016).





## 4.1 Humidity

On-board the ATR, humidity was measured by several instruments but the WVSS-II sensor was considered as a reference for
the calibration of the SAFIRE-CORE and SAFIRE-TURB datasets (Brilouet et al., 2021) because of its reliability and because
it was the least affected by the presence of condensation or very dry air (section 3.1.4). The Picarro CRDS measured water
vapor with a similar sampling, and its data were calibrated on the basis of laboratory measurements (section 3.6). During most
ATR flights, HALO (or, on Feb 11th, the P-3) was flying circles of 200 km diameter at high altitude, measuring water vapor
every 5 min and with a vertical resolution of about 10 m with Vaisala RD-41 dropsondes (George et al., 2021).

The comparison between these different measurements is presented in Fig. 16 for each ATR flight. For the SAFIRE-CORE
and Picarro measurements, the mean and standard deviation of the water mixing ratio are calculated over all the 'T-shortlegs'
segments (Table 2) associated with a given kind of segment. Note that the legs flown around the middle and the top of the
subcloud-layer have been considered together because the subcloud-layer is well mixed vertically (Albright et al, in prepara-
tion). For dropsondes, they are calculated over all available level-4 measurements in a layer comprised between the minimum
altitude minus 50 m and the maximum altitude plus 50 m sampled by the ATR for a given pattern. Most of the soundings
data within the EUREC$^4$A circle are derived from HALO dropsondes; these data include a correction for the dry bias of these
dropsondes (George et al., 2021).

For each flight and each pattern, the ATR measurements (SAFIRE-CORE and Picarro) generally exhibit a good agreement,
both in terms of mean humidity and standard deviation: over the cloud-base rectangles (R-patterns), the mean discrepancy
between the two datasets is 0.084 g kg$^{-1}$ (0.63 %) and 0.21 g kg$^{-1}$ (18.7 %) for the mean and standard deviation, respectively.
Thoses differences are slightly larger on L-patterns (0.27 g kg$^{-1}$ or 1.85 % and 0,21 g kg$^{-1}$ or 29,4 %, respectively) and on
S-patterns close to the surface (0.28 g kg$^{-1}$ or 1.86 % and 0.13 g kg$^{-1}$ or 33.5 %). The most notable exceptions occur on
Feb 11 2020 (RF17 and RF18), when the aircraft flew within or below precipitating clouds (Table 4): along a few legs, the
quality of CRDS measurements was affected by the presence of cloud droplets or precipitation in the air inflow system.

The ATR measurements are also in good agreement with the dropsondes data, including when all the measurements show
a small variability. Even the standard deviations show good agreement, which is somewhat surprising given the much larger
domain sampled by the HALO measurements. However, ATR and HALO measurements disagree more during RF09 (on
Feb 9th) and RF14 (on Feb 7th), when the cloud organization was very heterogeneous at the mesoscale (Table 3 reports a
prominence of flower and fish organisation on these days). During most of RF09 for instance, the ATR was sampling the clear-
sky area surrounding cloud flowers while the dropsondes were sampling both clear-sky and cloudy areas. During RF06 (on Jan
30), the variability of dropsonde measurements was very small because the corresponding HALO flight was much shorter and
was associated with the launch of only 2 dropsondes in the ATR area (as opposed to about 36 for other flights).

This comparison suggests that despite their different sampling and observing techniques, the ATR and HALO generally
measured statistically consistent variabilities of humidity around cloud-base, within the subcloud-layer, and near the surface.
The main discrepancies occurred when the scale of the cloud field organization was much larger than the scale of the area

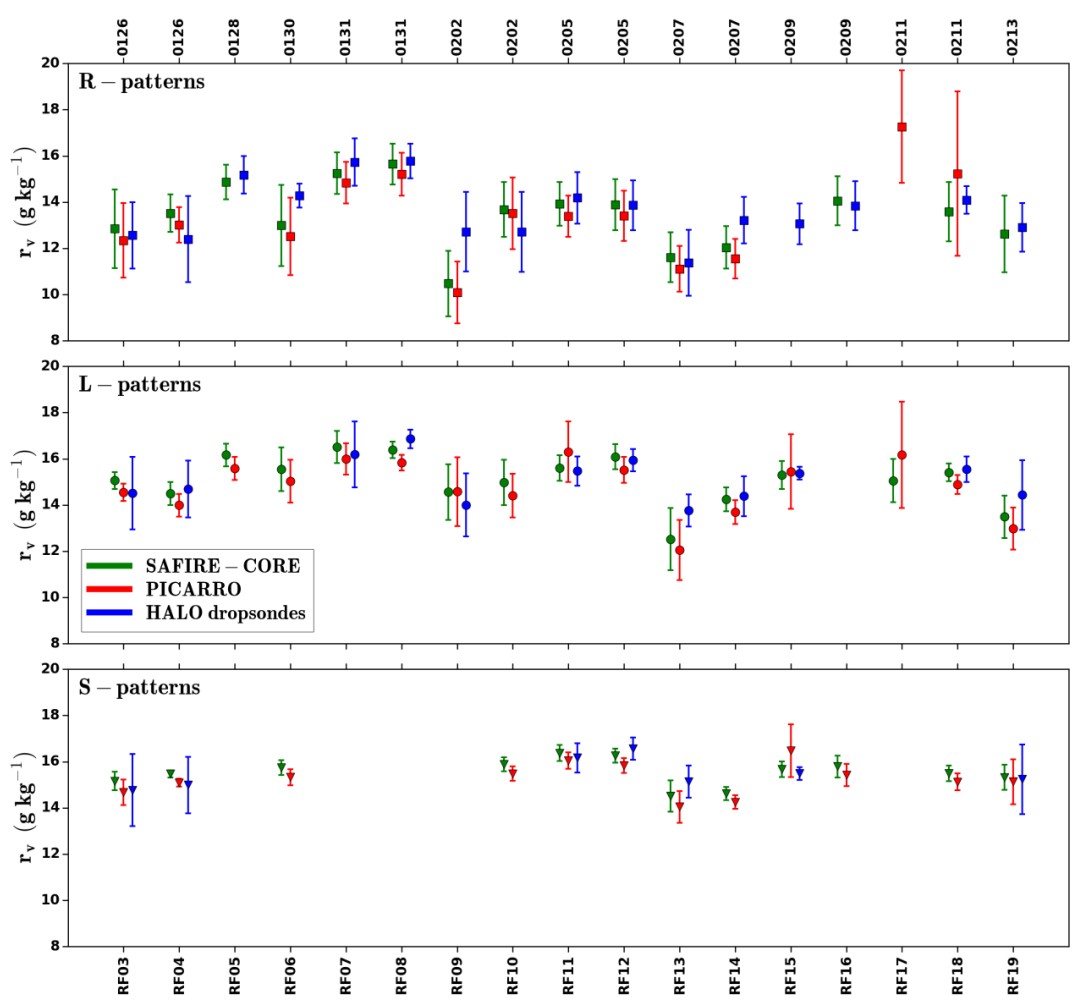

**Figure 16.** Comparison for each ATR flight of the water vapor mixing ratio inferred (green) from the SAFIRE-CORE dataset, (red) from the Picarro dataset and (blue) from the JOANNE dropsondes dataset. The top, middle and bottom panels are associated with different kinds of ATR patterns: R-patterns (flown at cloud base), L-patterns (flown in the subcloud-layer) and S-patterns flown near the sea surface, respectively. Note that the time axis is not linear. The standard deviation of dropsondes data is computed on the basis of all the individual dropsondes launched during each ATR flight.





probed by the ATR. In these cases, the differences are likely to be representative of real spatial differences associated with different samplings.

## 4.2 Horizontal wind

The wind was measured both in-situ using the aircraft probes (section 3.1.2) and through remote sensing using the Doppler
radars BASTA and RASTA (sections 3.5.2 and 3.5.4). In parallel, the wind was measured by dropsondes (George et al., 2021).

Fig. 17 compares the wind speed measured in the subcloud-layer or near the surface by the aircraft probes and by the HALO dropsondes at 60 m. The wind speed being quite similar near the surface and within the subcloud layer, we primarily consider the ATR measurements over L-patterns because these patterns are available for every flight and are associated with longer measurements. Despite the different flight patterns of the ATR and HALO, the average wind speed measured by the two aircraft
is very consistent: the dropsonde and ATR measurements correlate strongly (R = 0.98) and differ by only $0.7 \pm 0.5 \, \mathrm{m \, s^{-1}}$.

The two Doppler radars also measured the radial component of the horizontal wind along their line of sight. For BASTA, this information is retrieved on every flight, but for RASTA it is available only for RF03, RF04, and RF13 to RF19, when backward and zenith antennas were operating simultaneously. The radial component of the wind (perpendicular to the aircraft trajectory) derived from BASTA and RASTA along R-patterns is compared to the horizontal wind measured by the aircraft
probes and projected along the radars' line of sight (Fig. 17). For BASTA, the information is derived from the $3^{rd}$ gate (about 25 m from the aircraft) and for RASTA it is from the $4^{th}$ gate (about 60 m above the aircraft). The radar Doppler and in-situ measurements correlate strongly with each other (0.99 for the BASTA estimates, 0.94 for the RASTA estimates) and differ by $-0.61 \pm 1.22 \, \mathrm{m \, s^{-1}}$ and $0.24 \pm 1.3 \, \mathrm{m \, s^{-1}}$, respectively, over the whole campaign. The difference between BASTA estimates and in-situ measurements reduces to $0 \pm 1.22 \, \mathrm{m \, s^{-1}}$ when RF17 and RF18 are not considered, showing that most of the
uncertainty in wind retrieval from horizontal radar measurements occurs in the presence of rain. This bias arises because the vertical component of the wind has to be taken into account if the radar beams are not perfectly horizontal, which requires to account for the terminal speed of the hydrometeors. This might be corrected in future versions of the BASTA dataset. The radar Doppler measurements also agree with the HALO radiosonde measurements of the horizontal wind at the same height projected along the line of sight of the radars (not shown); on average over the campaign, the BASTA and RASTA estimates
differ from the HALO dropsondes data by -0.45 and +1 $\mathrm{m \, s^{-1}}$, respectively.

In summary, the horizontal wind measurements of the ATR are consistent with each other and with the dropsonde measurements made along the EUREC$^4$A circle. It confirms that the flight strategy resulted in a good statistical sampling of the lower-tropospheric wind.

## 4.3 Cloud-base cloud fraction

One of the original motivations for the EUREC$^4$A campaign was to test the mixing-desiccation mechanism by which increased convective mixing in the lower troposphere dries the atmosphere around cloud base and reduces cloudiness (Bony et al., 2017). This mechanism, which has been shown to contribute to the strong positive feedback of low-clouds and the high climate

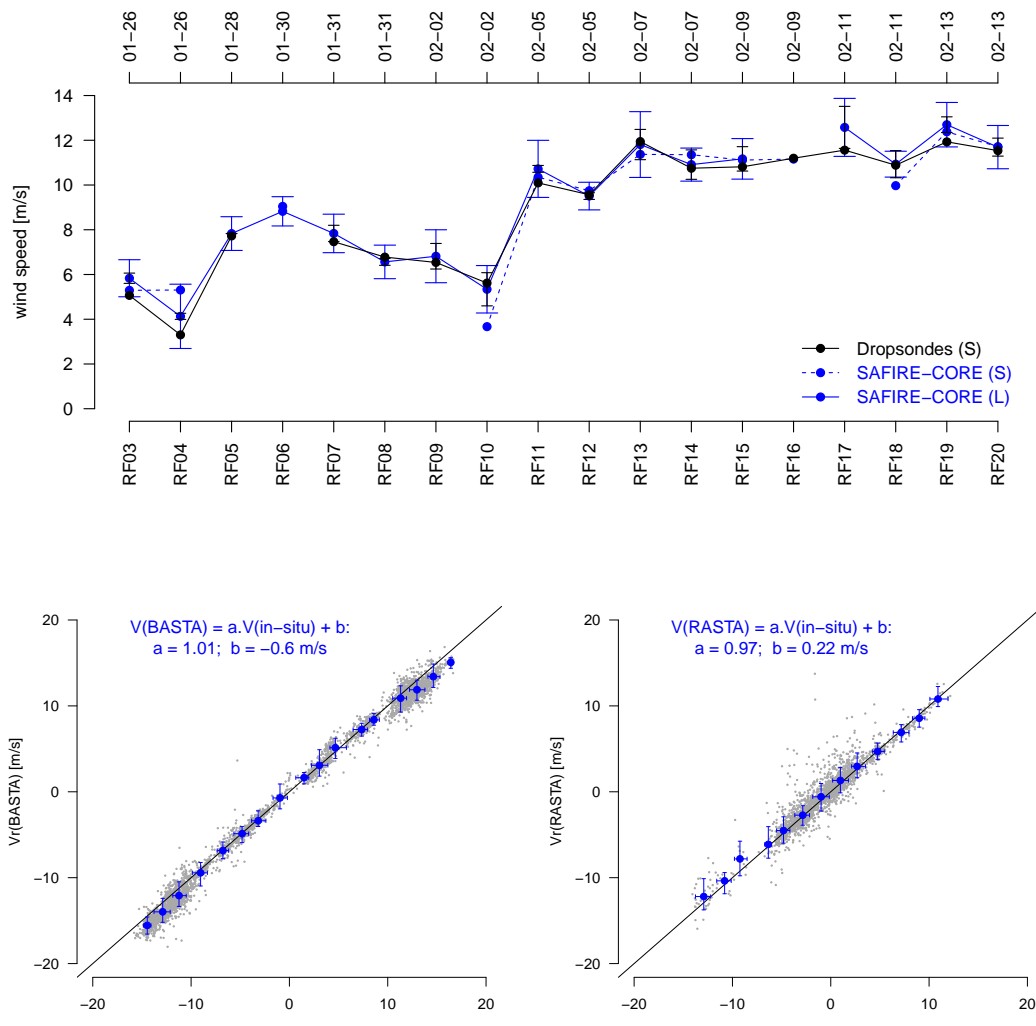

**Figure 17.** (Top) Comparison of the wind speed measured by the core instrumentation of the ATR (SAFIRE-CORE) in the subcloud layer with the near-surface wind measurements (at 60 m, close to the altitude of the near-surface legs of the ATR) from the HALO dropsondes. Note that the time axis is not linear. (bottom) Comparison of the radial component of the wind (perpendicular to the aircraft) derived from the aircraft probes (SAFIRE-CORE dataset) with that inferred from (left) the horizontally-staring BASTA Doppler radar at 25 m from the aircraft or (right) the vertically-pointing RASTA Doppler radar at 60 m above the aircraft; the comparison is done for all R-patterns flown at cloud-base during the EUREC⁴A campaign (note that there were no wind measurements from RASTA from RF05 to RF12 and on RF20). Data points are shown in grey, and the binned relationship between the two measurements is shown in blue.





sensitivity of a number of climate models, remains to be tested observationally. Such a test requires measuring the cloud fraction at cloud base $CF_b$ together with the lower-tropospheric mixing from convection and larger-scale vertical motions.

Reflecting the view that clouds are both bodies interacting with radiation, collections of particles, and a particular state of atmospheric water (Siebesma et al., 2020), we estimate $CF_b$ in different ways, using various observations ranging from lidar and radar remote sensing, to in-situ microphysical measurements (defining the cloud mask either from the cloud particle properties directly or from the equivalent radar reflectivity calculated from these properties), to high-frequency humidity measurements.

Using horizontal lidar-radar measurements from ALiAS and BASTA together with the BASTALIAS cloud detection algo-
rithm described in section 3.5.3, we diagnose the cloud fraction within the rectangle area associated with the R-patterns flown at cloud base: for each R-pattern, we divide the total number of points classified as 'cloudy' along the instruments' line of sight by the total number of points where a cloud detection is possible. The resulting time series of $CF_b$ is shown in Fig. 18, which uses a reflectivity threshold for the cloud-drizzle transition of -20 dBZ. Using a different threshold (-17 dBZ or -15 dBZ) makes very little difference in the time series (not shown). $CF_b$ is small on average (3.5 %) but it ranges from 0 to 6 % across
flights. Minima in $CF_b$ occurred during RF09 and RF14, when the ATR was flying within the clear-sky area of a field of 'cloud flowers' organised at the mesoscale (Table 4).

Most of the cloud fraction (from 60 to 100 %) is composed of clouds which were detected by the lidar only (Fig. 18). As explained in section 3.5.3, it is because a large proportion of clouds in the trades are small (a few hundred meters) and optically thin (especially at cloud base where the liquid water content is small), and because in the trade-wind boundary layer, horizontal
radar measurements can only detect clouds over a range of 2-3 km from the aircraft, while horizontal lidar measurements can detect clouds over a distance at least twice larger.

Using reflectivities at the 5th gate (i.e. about 90 m above the aircraft), and a threshold of -20 dBZ to distinguish clouds from drizzle, a cloud fraction can also be diagnosed from the vertically-pointing cloud radar measurements (RASTA, section 3.5.4). The $CF_b$ estimates from RASTA are in good agreement with those from BASTALIAS, except on RF17 and RF19 during which
RASTA measurements seem to be dominated by rain.

A cloud fraction estimate can also be diagnosed from in-situ measurements of cloud microphysics (section 3.4.2) using two methods. The first one consists in using the PMA hydrometeors masks defined as LWC $\geq 0.010$ g m$^{-3}$ and D $< 100$ $\mu$m, and the drizzle mask as $100 \leq$ D $< 500$ $\mu$m. We then compute a cloud fraction along the aircraft trajectory as the ratio between the number of points classified as 'cloud-only' over the total number of valid measurements. Although the sampling along
the aircraft trajectory is much less extensive than that of horizontal lidar-radar measurements, the time evolution of the $CF_b$ derived from PMA data is highly correlated (R = 0.90) with that from BASTALIAS (Fig. 18). The presence of drizzle can make the definition of the cloud base cloud fraction ambiguous as drizzle can fall within the cloud base. For this reason, we also compute a cloud fraction considering both clouds and drizzle, diagnosed either from BASTALIAS data or from PMA cloud microphysical data. The cloud+drizzle fraction differs from the cloud-only fraction mostly on Feb 11, during which the ATR
flew during night time (RF17) or in the morning (RF18) in the presence of cloud flowers and a strong ascending motion in the free troposphere (Table 2.5). However, even in the presence of drizzle and rain, estimates from the two measurements are very consistent with each other (R = 0.90, Fig. 18).

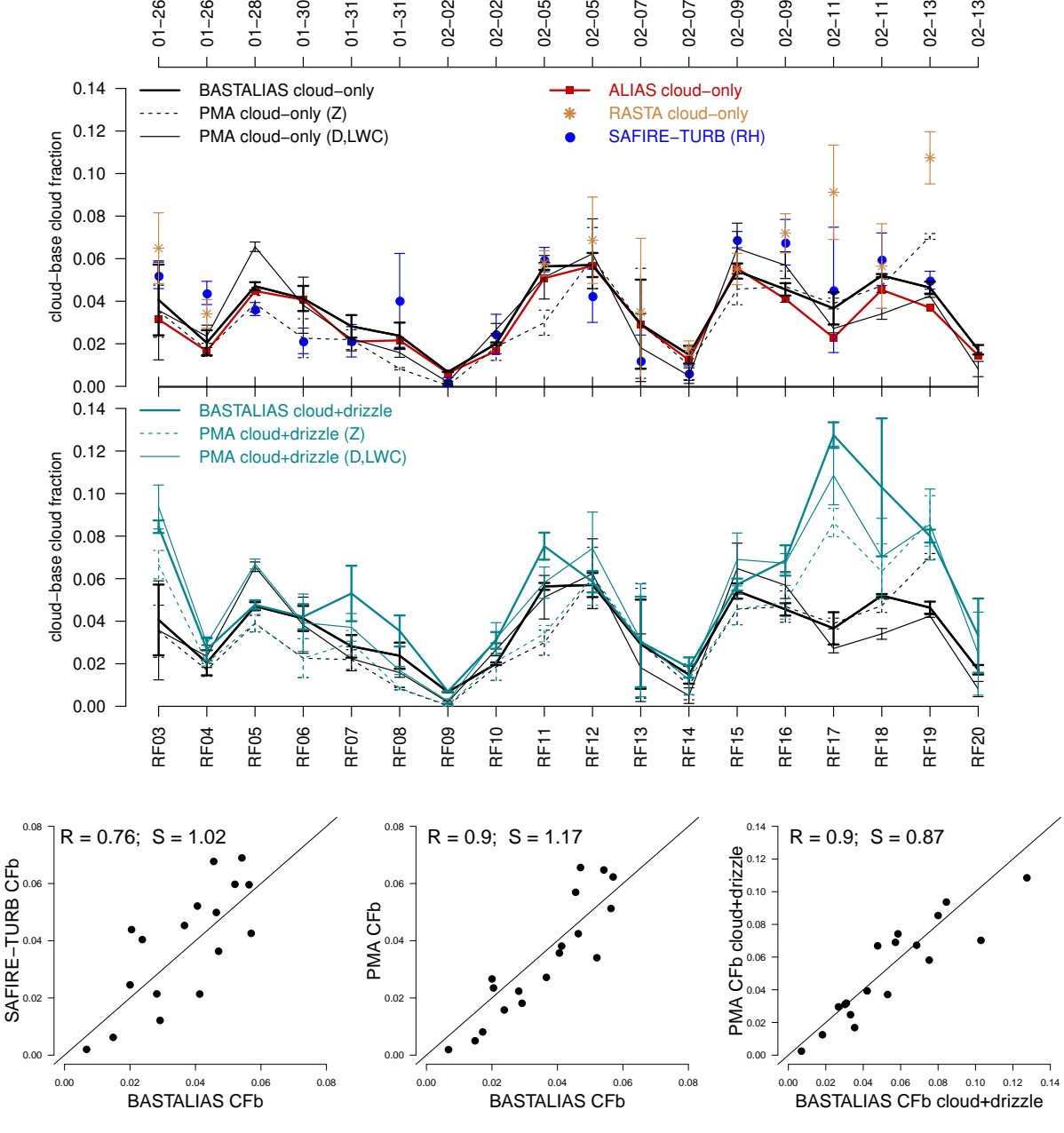

**Figure 18.** (top) Cloud-base cloud fraction (CF$_b$) inferred from ATR observations across or along the R-patterns flown at cloud-base: from horizontal radar-lidar remote sensing (BASTALIAS), from horizontal lidar only (ALiAS), from in-situ microphysical measurements (PMA) using an hydrometeor classification based either on (D, LWC) or on reflectivities (Z) inferred from the particle size distribution, from in-situ humidity measurements at the turbulence scale (RH from SAFIRE-TURB), and from the vertically-pointing RASTA radar. (middle) Comparison of the fractional area covered by "clouds only" (in black) or by "clouds+drizzle" (in orange) inferred either from the BASTALIAS lidar-radar dataset or from the PMA dataset using both types of hydrometeor classification. Note that the time axis is not linear. (bottom) Correlation (R) and linear regression coefficient (S) between the CF$_b$ estimates derived from different ATR datasets.



Finally, recognizing that clouds occur in saturated (or, in the presence of sea salt, nearly saturated with respect to pure water saturation) conditions, it is possible to define a pseudo cloud fraction from the high-frequency (25 Hz) / small-scale

(4 m) measurements of relative humidity: using the SAFIRE-TURB dataset, we reconstruct high-resolution timeseries of humidity mixing ratio, temperature and then relative humidity by adding the turbulent fluctuations of each variable measured over stabilized segments (T-shortlegs) to the mean of each segment (section 3.3, (Brilouet et al., 2021). Then, by counting the proportion of measurements having a relative humidity exceeding a threshold $RH_c$, we define a pseudo cloud-fraction from the SAFIRE-TURB dataset. The time series of $CF_b$ obtained with this method using a threshold $RH_c = 0.98$ is in good

agreement and correlates well with the cloud fraction estimated from BASTALIAS (R = 0.76) or PMA data (R = 0.71). The best correlation with BASTALIAS and PMA data is obtained during the second half of the campaign (after RF09), when the high-frequency measurements of humidity were of best quality (Brilouet et al., 2021).

The values of $CF_b$ derived from the different datasets obviously depends to some extent on the thresholds used to define cloudy conditions. However, sensitivity tests suggest that the high correlation among the different estimates remains for a range

of threshold values. Considering the diversity of measurement techniques (in-situ microphysical and turbulent measurements, horizontal lidar-radar measurements, vertical radar measurements) and spatial samplings (rectangle perimeter or rectangle area) leading to consistent results, the $CF_b$ estimates from the ATR can be considered as robust.

### 4.4   Water isotopic composition

To assess the consistency of isotopic data between the ATR Picarro dataset and the ground-based measurements from the BCO

(Fig. 19), we select the data collected at altitudes lower than 400 m (this height is well below cloud base for all flights and contains the ground-based measurements at the airport) and compute the mean value for each flight ($ATR_{alt \leq 400m}$). The BCO data is averaged over each flight's period ($BCO_{flights}$). The vapor sampled by the Picarro onboard the ATR was drier (-0.7 g kg$^{-1}$) and more depleted (-1 ‰ in $\delta^{18}O$ ‰ and -4.3 ‰ in $\delta^2H$) than at the BCO (Table 10). The d-excess was nearly identical except for RF04, RF07 and RF08 for which the d-excess at the BCO was lower. Similar differences between the BCO and the

R/V Atalante were recorded during a comparison stop offshore the BCO (personal communication Gilles Reverdin). A possible explanation for the observed differences could be the effect of sea spray evaporation due to the wave activity at the cliff in front of the BCO that enriches and moistens the air close to the land-based site. The flight-to-flight isotope variability recorded by the ATR agrees well with the one observed at the BCO: correlations between $ATR_{alt \leq 400m}$ and $BCO_{flights}$ range from 0.7 to 0.82 (Table 10). Due to their spatial separation, the instruments did not measure the exact same air, or if so, due to advection,

with a time lag. Therefore, the qualitative and quantitative match between the datasets suggests that the measurement are of good quality, and that distinct mesoscale environmental conditions were measurable during the different research flights.

### 5   Data availability

All the datasets discussed in this paper are available on the EUREC[4]A database of AERIS (https://eurec4a.aeris-data.fr), and their respective DOIs are summarized in Table 11.





|  | $\delta^{18}$O | $\delta^2$H | d-excess | Specific humidity |
|---|---|---|---|---|
|  | [‰] | [‰] | [‰] | [g kg$^{-1}$] |
| ATR$_{alt \leq 400m}$ | -10.6 ± 0.6 | -71.0 ± 2.2 | 14.1 ± 2.7 | 15.2 ± 0.7 |
| BCO$_{flights}$ | -9.6 ± 0.7 | -66.7 ± 2.4 | 10.0 ± 3.3 | 15.9 ± 0.7 |
| Correlation | 0.75 | 0.80 | 0.70 | 0.82 |

**Table 10.** Mean values and standard deviations over all flights for the measurements made by the Picarro onboard the ATR and at the BCO during the ATR flights, as well as the Pearson correlation between the two data sets.

| Dataset | Link | Description | Principal investigator | Citation |
|---|---|---|---|---|
| Flight segments | doi.org/10.25326/315 | This paper | S. Bony | Bony et al. (2021) |
| Satellite movies | doi.org/10.25326/299 | This paper | B. Fildier | Fildier et al. (2021b) |
| SAFIRE-CORE | doi.org/10.25326/298 | This paper | CNRM/TRAMM | CNRM/TRAMM et al. (2021) |
| SAFIRE-RADIATION | doi.org/10.25326/84 | This paper | CNRM/TRAMM | CNRM/TRAMM (2020a) |
| SAFIRE-CLIMAT | doi.org/10.25326/61 | This paper | CNRM/TRAMM, LOA | CNRM/TRAMM (2020b) |
| SAFIRE-CAMERA | doi.org/10.25326/297 | This paper | C. Cornet | Cornet and JIANG (2021) |
| PMA/CDP-2 | doi.org/10.25326/209 | This paper | P. Coutris | Coutris and Schwarzenboeck (2021a) |
| PMA/2D-S | doi.org/10.25326/219 | This paper | P. Coutris | Coutris and Schwarzenboeck (2021b) |
| PMA/CloudComposite | doi.org/10.25326/237 | This paper | P. Coutris | Coutris (2021) |
| UHSAS | doi.org/10.25326/220 | This paper | P. Coutris, G. Ehses | Coutris and Ehses (2021) |
| BASTA | doi.org/10.25326/314 | This paper | J. Delanoë | Le Gac et al. (2021) |
| BASTALIAS | doi.org/10.25326/316 | This paper | J. Delanoë | Chazette et al. (2021) |
| RASTA | doi.org/10.25326/313 | This paper | J. Delanoë | Caudoux et al. (2021) |
| PICARRO | doi.org/10.25326/244 | Bailey et al. (submitted) | F. Aemisegger | Aemisegger et al. (2021a) |
| SAFIRE-TURB | doi.org/10.25326/128 | Brilouet et al. (2021) | M. Lothon, P.-E. Brilouet | Lothon and Brilouet (2020) |
| ALIAS basic data | doi.org/10.25326/57 | Chazette et al. (2020) | P. Chazette | Chazette (2020a) |
| ALIAS cloud products | doi.org/10.25326/58 | Chazette et al. (2020) | P. Chazette | Chazette (2020b) |
| ALIAS aerosol products | doi.org/10.25326/59 | Chazette et al. (2020) | P. Chazette | Chazette (2020c) |

**Table 11.** List of ATR datasets derived from EUREC$^4$A.

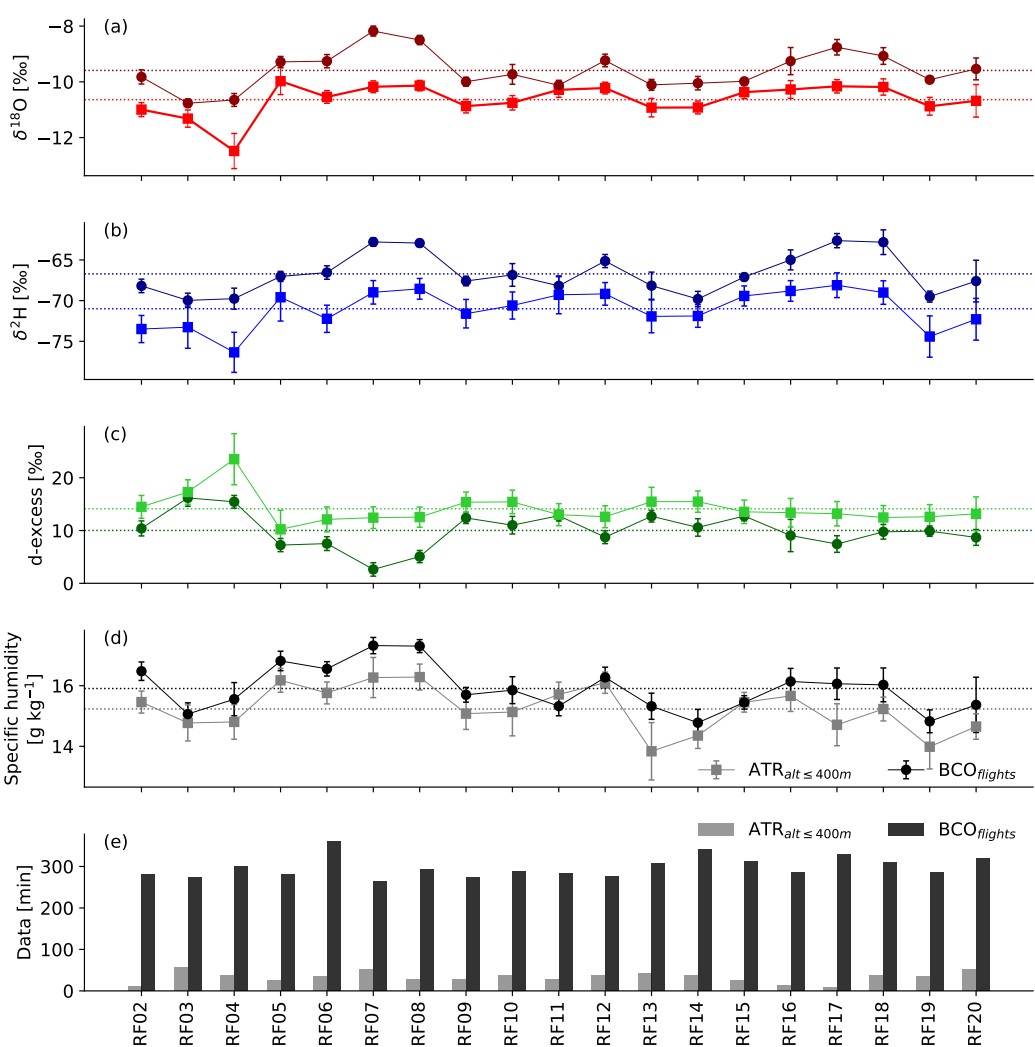

**Figure 19.** Comparison of the ATR isotope measurements below 400 m ($\delta^{18}$O, $\delta^2$H and d-excess expressed in ‰, specific humidity in g kg$^{-1}$, and length of measurements expressed in min) with the ground-based measurements from the Barbados Cloud Observatory (BCO) during each ATR flight.



## 6  Summary and conclusions

The EUREC$^4$A field campaign, which aims at better understanding the link between clouds and circulation in the region of the trades, based its core experimental strategy on the coordinated operations of two research aircraft (Bony et al., 2017; Stevens et al., 2021): the French ATR aircraft flying in the lower troposphere and the German HALO aircraft flying at an altitude of 9-10 km. This paper presents the EUREC$^4$A's ATR operations and presents the 19 ATR flights (totaling approximately 82 flight

hours) that took place from Jan 25 to Feb 13, 2020 over the tropical Atlantic ocean, east of Barbados.

The ATR mission focused on characterizing the thermodynamic, dynamical, microphysical, turbulent and cloud properties of the lower atmosphere. One of its specific roles was to measure the cloud-fraction around cloud-base to help test low-cloud feedback mechanisms. For this and other purposes, the ATR was equipped with a rich and extensive instrumentation composed of in-situ sensors, radiometers and active remote sensing. Eighteen coordinated research missions followed a repeated flight

plan consisting of rectangles (or R-patterns) flown at cloud-base or cloud-top, L-legs flown within the subcloud layer (L-patterns), straight legs flown 60 m above the sea surface (S-patterns), and ferry legs flown in the lower free troposphere above clouds.

The first part of this paper presents the ATR operations, the flight patterns and the flight segmentation (summarized in a collection of YAML files). It also shows that during its 19 missions, the ATR sampled very contrasted environmental conditions.

The second part of this paper presents the ATR instrumentation used during EUREC$^4$A: 3 temperature sensors, 5 humidity sensors, 2 broadband radiometers, an infrared spectrometer, 2 visible cameras, 6 microphysical probes, an horizontally-staring backscatter lidar, 2 Doppler cloud radars (1 pointing horizontally and 1 pointing vertically), and a laser spectrometer for water isotopologues. The paper presents the different instruments, highlighting the complementarity that results from their different working principles. Then it presents the main aspects of the data processing and the datasets produced from the different

measurements: the core thermodynamical, dynamical and radiative measurements (SAFIRE-CORE, SAFIRE-RADIATION, SAFIRE-CLIMAT, SAFIRE-CAMERA), turbulence measurements (SAFIRE-TURB, Brilouet et al. (2021)), aerosol (UHSAS) and cloud microphysical (PMA) measurements, horizontal lidar measurements (ALiAS, Chazette et al. (2020)), horizontal radar (BASTA) and lidar-radar (BASTALIAS) measurements, vertical radar measurements (RASTA) and water isotopic measurements (Picarro).

Finally, the paper assesses the consistency among the different ATR measurements, and between the ATR measurements and those performed by other instruments on different platforms such as HALO or the BCO.

The large variability of the aerosol load in the atmosphere (ranging from 50 to 500 cm$^{-3}$) is measured consistently by the ALiAS lidar and by UHSAS microphysical probes (sections 3.4.1 and 3.5.1). The measurements of humidity, wind and cloud-base cloud fraction also exhibit a good consistency among the different ATR datasets. The mean specific humidity measured

by in-situ sensors differs from that measured by the ATR Picarro laser spectrometer by less than 0.1 g kg$^{-1}$ at cloud-base, and by less than 0.3 g kg$^{-1}$ within the subcloud layer; larger disagreements occur on RF17 and RF18 when the Picarro measurements were impacted by cloud droplets and precipitation in the air inflow system. Estimates of the radial component of the wind from the Doppler cloud radars are in good agreement with the aircraft probes measurements, with a discrepancy of -





0.61 ± 1.22 m s⁻¹ for BASTA and 0.24 ± 1.3 m s⁻¹ for RASTA. Finally, a cloud-base cloud fraction was estimated for the first

time from horizontal lidar-radar measurements (BASTALIAS), and other estimates were derived from in-situ microphysical or turbulent measurements along the aircraft trajectory. These different estimates are in good agreement with each other (correlations of 0.76 and 0.91 between BASTALIAS and SAFIRE-TURB or PMA estimates, respectively). The cloud-base fractional areas associated with "clouds plus drizzle" estimated either from BASTALIAS or from PMA datasets are also consistent with each other (correlation of 0.92). The good consistency obtained despite fundamentally different measurement techniques and

different atmospheric samplings (in-situ sensors sample the atmosphere along the aircraft line of flight, i. e. along the rectangle perimeter, while horizontal lidar-radar remote sensing samples the interior of the rectangle) shows that the ATR measurements of humidity, wind and cloud-base cloud fraction are robust.

The ATR measurements of humidity and winds exhibit a good consistency with HALO dropsondes measurements: the wind measurements from the ATR and from HALO are highly correlated (R = 0.98) and differ by only 0.7 ± 0.5 m s⁻¹; the ATR

humidity measurements are also in good agreement with the dropsondes data, both in terms of mean and variability. It shows that the measurements made by HALO and the ATR are consistently representative of the explored area, despite the complexity of the cloud organization and its inner heterogeneity.

These results thus verify two premises which were at the basis of the EUREC⁴A experimental strategy: 1) it is possible to measure the cloud-base cloud fraction in a robust way, and 2) the repeated flight patterns of HALO and the ATR allow us to

sample the atmosphere statistically in a consistent way, except when the cloud field is organized on a scale much larger than the scale of the ATR flight pattern (which only occurred twice out of the 18 flights). It is therefore legitimate to use observations from the different EUREC⁴A platforms together to carry out process studies. The availability of data from the ATR and other platforms together with the large diversity of environmental conditions and clouds encountered during the campaign should thus make it possible to better understand the physical processes underlying the cloud-circulation interactions in the trades.

**Appendix A: Satellite movies**

Satellite animations were made to visualize the clouds scenes sampled by the ATR and other platforms during the campaign. Geostationary images shown here are from the GOES-16 visible channel in daytime (channel 2) and infrared channel in nighttime (channel 13). They are retrieved at 1 min time increment and 500 m resolution in daytime, 2 km resolution in nighttime. The code is modular (Fildier et al., 2021a), and it can be used to generate movies with any combination of platforms

making measurements in the region captured by the mesoscans (50.1323-61.3010°W, 8.2457-17.9343°N). It can also include the trajectory of the radiosondes and dropsondes launched. Here, movies are restricted to the ATR track in the domain 57-60°W, 11.8-14.8°N. The source code to generate GOES movies is archived at https://doi.org/10.5281/zenodo.4777954 and in the github repository https://github.com/bfildier/EUREC4A_movies.





**Figure B1.** Vertical trajectories of each ATR flight, with the main patterns highlighed in color. R-patterns at cloud base and/or cloud top (orange), L-patterns in the subcloud-layer (blue), S-patterns near the sea surface (red), Ferry legs (black) and upward/downward profiles (turquoise). Also reported (dashed line) is the subcloud layer top height diagnosed from dropsondes (Table 3).

## Appendix B: Research flight trajectories

**Appendix C: ATR instrumental configuration**



**Figure B1.** (continued)

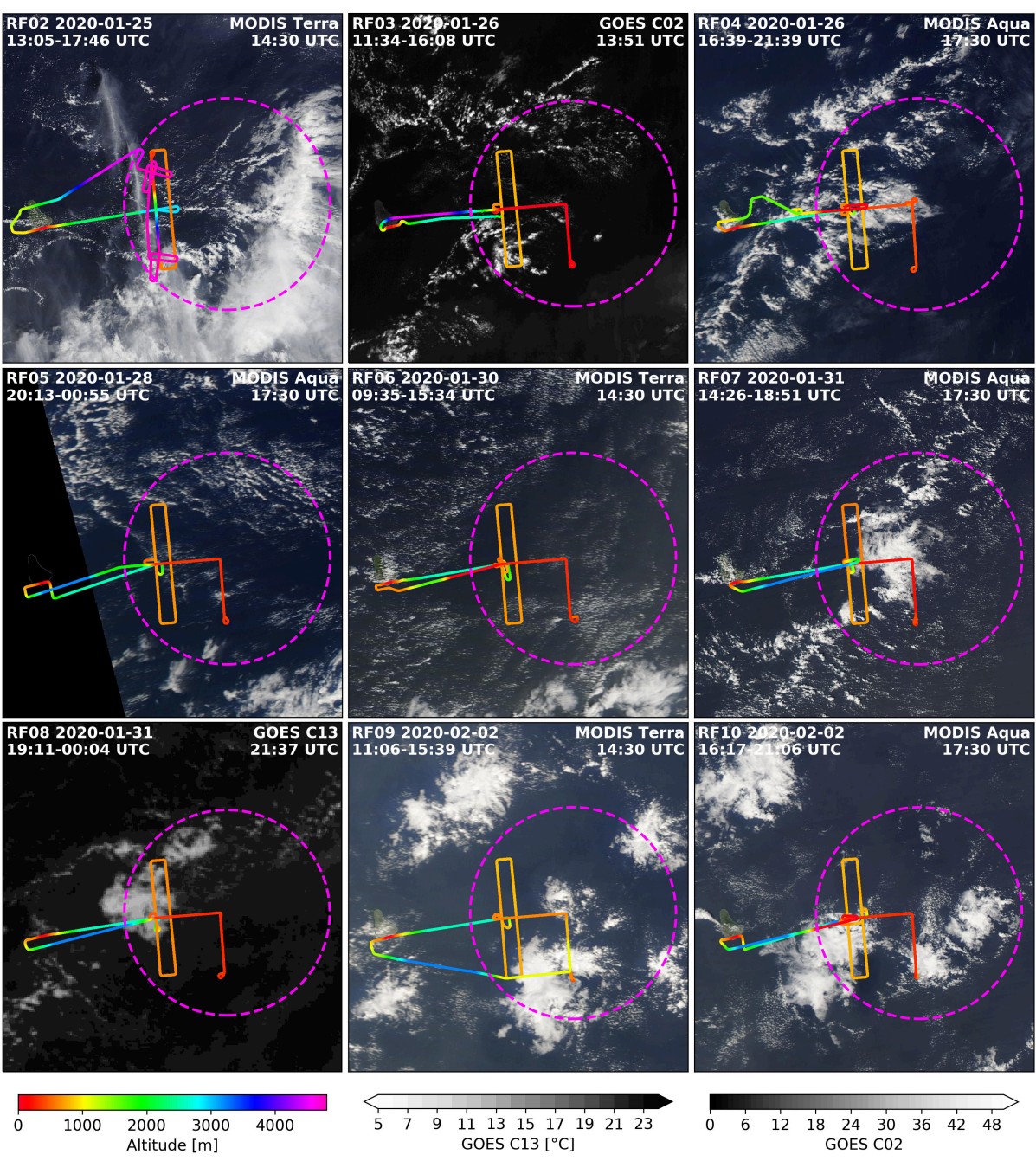

**Figure B2.** Longitude-latitude trajectories of the ATR coloured by the flight altitude. For repetitive flight patterns (e.g., the rectangles), only the last repetition is visible due to the overlap. The dashed circle shows the EUREC[4]A circle along which HALO was flying. The ATR tracks are shown on top of a satellite snapshot of the domain (57-60°W, 11.8-14.8°N) derived either from MODIS Terra /Aqua images (when available during the flight), or from GOES-16 (using either the C02/visible or C13/infrared channel) at about mid-flight time.





**Figure B2.** (continued)



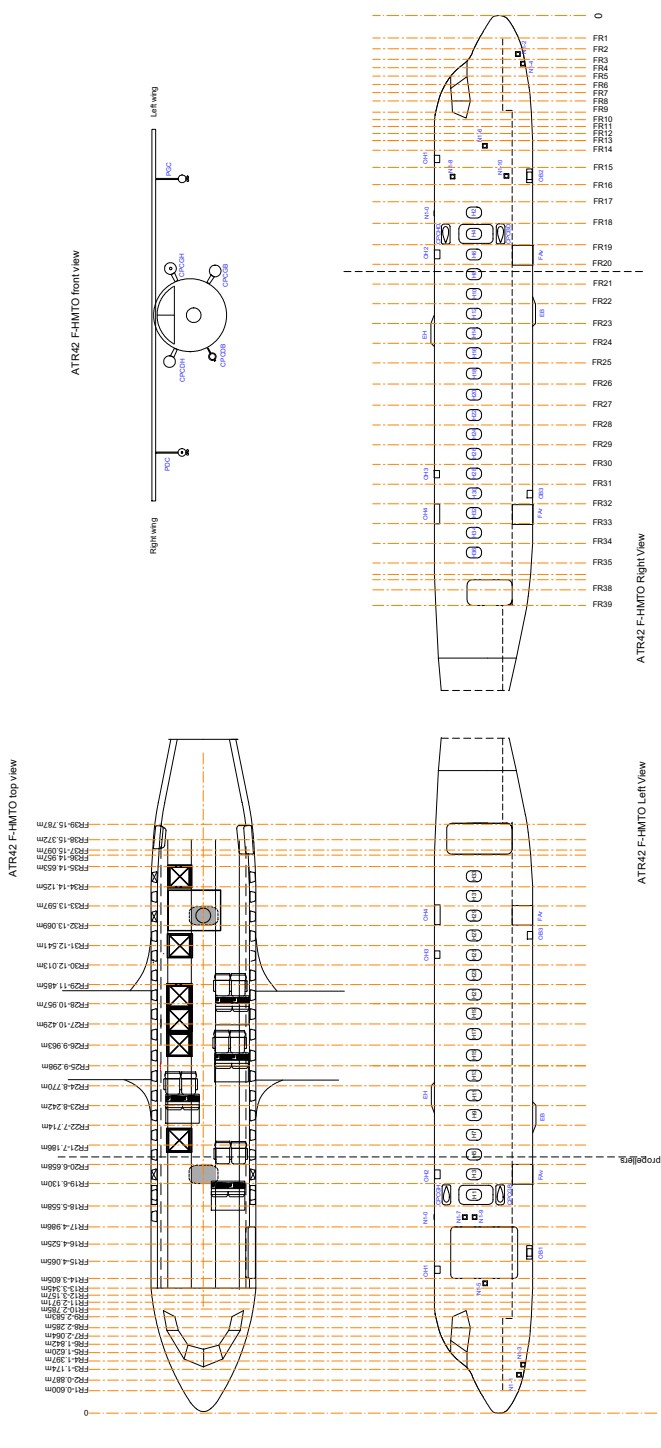

**Figure C1.** Instrumental configuration of the ATR showing the nomenclature used in Tables 5 to 9.



*Author contributions.* The people responsible for the processing of the different ATR datasets are listed in Table 11. SB led the ATR team and coordinated the scientific operations with ML, JD and BS. JCC and JPD led the SAFIRE operations with the support of AB. PC, CF, JT and AB were responsible for the lidar measurements; JD, CLG and CC for the radar measurements; ML, PEB and SAFIRE/TRAMM (JCE and PR) for the core and turbulence measurements; AS, PC and CG for the microphysical measurements; FA and LV for the isotopic
measurements, and CC and TJ for the camera measurements. RV was responsible for the ground support with the help of JV and NR. TA, HB, MC, LG, TJ, CL, JL, TP, FP, KS and GV were the technicians and engineers who took care of the integration and the operation of the instruments on-board the ATR. JFB, GS, DD and CL flew the ATR. BF and LTP produced the GOES animations with the help of Hauke Schulz. Fig. 2a and Fig. B2 were prepared by LV, Fig. 3 by RV, Fig. 2b, Fig. 7 and Fig. C1 by JCE and LG, Fig. 8 by CC, Fig. 12 and Fig. 15 by JD, Fig. 16 by PEB, ALA and FA, and Fig. 19 by FA. Tables 3 and 4 have been prepared by JV, and Table 10 by LV. SB prepared
all other Tables and Figures, analyzed the consistency among the different ATR datasets and wrote the paper with ML, JCE, PC and JD, with contributions from FA, RV and JV. All authors edited the manuscript. SB and BS conceived the strategy and led the EUREC$^4$A field campaign.

*Competing interests.* The authors declare having no conflict of interest.

*Disclaimer.* This article is part of the special issue 'Elucidating the role of clouds-circulation coupling in climate: datasets from the 2020
(EUREC$^4$A) field campaign". It is not associated with a conference.

*Acknowledgements.* The Caribbean Regional Security System (RSS), especially Major George Harris and Captain Calvert Herbert, are gratefully acknowledged for hosting the ATR and the ATR team in Barbados in the best conditions during the EUREC$^4$A experiment. The authors thank David Farrell, Principal of the Caribbean Institute for Meteorology and Hydrology (CIMH) and Captain Don Chee-A-Tow, Honorary Consul of France in Barbados, for their precious support during the preparation and the realization of the campaign. The
Department of Civil Aviation in Barbados, Andrea Hausold from DLR, the British Twin Otter team, the German HALO team, the NOAA P3 team and the Boreal team are also gratefully acknowledged for their help, constructive spirits and cooperation during the airborne operations. The authors are grateful to Caroline Muller, Heike Kalesse, Katharina Baier, Johannes Röttenbacher, Friedhelm Jansen, Sabrina Schnitt, Geet George, Martin Wirth, Silke Gross and Stefan Kinne for their help of the ground support during the ATR operations, and to Vincent Douet, Karim Ramage and AERIS, for their help in distributing the ATR datasets on the EUREC$^4$A database.

*Financial support.* This project was funded by the European Research Council (ERC) under the European Union's Horizon 2020 research and innovation programme (EUREC$^4$A Advanced grant No 694768). It also received support from the French National Center for Space Studies (CNES) through the EECLAT proposal and from Meteo-France. LV and FA acknowledge funding from the Swiss National Science Foundation Grant No 188731.



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
