# Peer review of "EUREC4A observations from the SAFIRE ATR42 aircraft"

_Earth System Science Data, 2021_

## Author Comment (AC1)

**RESPONSE TO THE REVIEWS (MANUSCRIPT ESSD-2021-459)**

S. BONY ET AL.

We thank Alan Blyth and the second Reviewer for their thoughtful and constructive comments on our manuscript.

Below is an itemized response to the different issues raised in the reviews (Reviewers' comments are in blue, our responses are in black). The revised version of the manuscript addresses all comments and suggestions.

In addition, Figures 16 and 19 have been revised so that their style is now consistent with that of other figures (no change in the plotted quantities), and the EIS column of Table 3 was removed because it was a bit redundant with the LTS column (both are measures of the lower tropospheric stability).

**Point-by-point response to Alan Blyth's comments (Reviewer 1)**

This paper describes the observations gathered by the SAFIRE ATR42 research aircraft during the EUREC4A field campaign. The paper is very clear, well written, and the figures are well chosen. I enjoyed reading it. It provides a very valuable and informative summary of the ATR aircraft operations and description of the datasets produced. The section on the consistency among observations is particularly interesting and important for the EUREC4A community. The paper will be a excellent resource for scientists wishing to analyse SAFIRE ATR42 EUREC4A data now, and for many years to come. A few relatively minor comments follow which the authors may wish to consider.

l205 I wondered what the actual description should be. The beginning and end of these episodes is a bit unclear in reality and also the second episode of elevated aerosols was over a few days, not just on 11 Feb. As far as I understand, there were two periods of mineral dust, with the second period containing significant biomass burning. I would suggest 30 Jan-6 Feb and 9-12 Feb based on data from Ragged Point (Peter Gallimore, personal communication), but it could be +/1 day. Perhaps use the word "about"? We now write: *Finally, episodes of dust occurred from about Jan 31 to Feb 5, and on Feb 11 (Table 4), consistent with those observed on 30 Jan - 6 Feb and on 9-12 Feb at Ragged Point in Barbados (Peter Gallimore, personal communication), from the R/V Ron Brown (Stevens et al, 2021) and in atmospheric composition reanalyses (Chazette et al. 2022).*

Section 3.1.3. It would be good to consider Lawson and Cooper (1990, JAOT, 7, p480), particularly with regard to the wetting caused by cloud drops. We now write: *Despite its housing, the response time of the Rosemount sensor can sometimes be affected by the presence of cloud droplets (Lawson and Cooper, 1990).*

l265. Is it correct that *both* the Rosemount and fine wire temperature data are processed at 1 Hz and at 25 Hz? Just the Rosemount data were processed at 1 Hz and at 25 Hz. The sentence has been corrected accordingly.

l285. Perhaps "sensors mentioned above"? Done.

l285. Reference at this point for the WVSS-II? Similarly for the Licor and KH20, and perhaps other instruments mentioned? Or collectively for a previous project? We now cite Fleming and May (2004) for the WVSS-II, plus Smit et al. (2014) and Vance et al. (2015) regarding the assessment of this sensor on an aircraft. For the Licor, we now cite an internal internship report about the adaptation of the instrument to the ATR (Rozen and Muskardin, 2007) and a comparison made by Lampert et al. (2018) between the airborne-Licor and a reference fast sensor (Lyman-alpha, Friehe et al 1986) which is not manufactured anymore. For the KH20, we now cite Campbell et al. (1985) and Foken and Falke (2012), we mention a previous adaptation of the sensor to an aircraft (Kotani and Sugita, 2004), and we cite an internal internship report about its adaptation to the ATR (Charoy 2015).

l314. g m$^{-3}$ Corrected.

l321. Add "respectively"? Done.

l415. ...subcloud layer and out of cloud at the cloud base level? Just to be clear? Likewise, in the next line, ... at the cloud-base level? Done.

l417. Refer back to the last sentence of Section 2. Done. We now refer to Table 4 and section 2.5.

l430. References for the CDP-2 and 2D-S when they are first mentioned. Done.

l474. It be would be useful to discuss the overlap size range for the two probes. Was there always good agreement that allowed there to be a unique distribution to be created? We added:

*To do so, the 2D-S size distributions are interpolated to match the CDP-2 bin resolution on the 10 to 50 μm overlap region. When data from both probes are available, we use CDP-2 data up to 31 (± 1) μm, between 33 and 43 (± 1) μm, we use the average of CDP-2 and 2D-S concentrations, and beyond 50 (± 5) μm, we use 2D-S data . When CDP-2 data are not available (data are set to NaN values whenever a probe does not operate), the first two bins of the 2D-S are omitted such that the composite spectra start at 30 (± 10) μm. Note that a ± 1 second offset was added to the 2D-S data whenever it improved the correlation between the LWC retrieved from the CDP and 2D-S data in the 25-45 μm overlap region.*

*As the CDP and 2D-S were mounted on two different wings about 10-15m apart (Table 8), it could happen that only one of the two wings crossed a cloud, thus generating some inconsistency between the measurements of the two probes. Therefore, the composite product comes with a variable (compo_index) that describes the composite and qualifies the overlap between the two sondes (1: CDP data only, 2: 2D-S data only, 3: CDP and 2D-S in good agreeement, and 4: CDP and 2D-S in poor agreement). In the*

*future, inconsistencies could be limited by producing another composite that would use the 2-50 µm measurements from the FCDP probe (rather than the CDP), as this probe was mounted just below the 2D-S.*

*From the composite size distributions we calculate microphysical quantities such as the total concentration ($N_T$), liquid water content (LWC, third moment of the distribution assuming a density of 1g cm$^{-3}$), median volume diameter (MVD, defined as the median of the cumulative mass size distribution), and a series of masks that indicate the presence of cloud, drizzle or rain drops.*

l486. Semi-colon after Table 8. Done.

l491. To be consistent, perhaps ... underestimate the LWC measurement when such large drops are present. Drizzle is defined in the previous paragraph. I do wonder, for cloud lwc, how much of an error there will be due to this incomplete evaporation for such large drops which occur in relatively low concentrations. Is it larger than the error due to mis-sizing of large drops? Done.

l497. It might be better to say the concentration of cloud drops is disproportionally less in a few cases with larger aerosol concentrations. There are a few points where that is not true, which might suggest differences in hygroscopicity as mentioned. We made this change (*At larger aerosol concentrations, the cloud droplet concentrations are disproportionally less in cases with larger aerosol concentrations.*), and added: *However, such a sublinear relationship between CCN and cloud drop concentration is not uncommon and different interpretations have been proposed for this feature, including a measurement artifact known as 'coincidence' (Lance, 2012). Since the CDP-2 probe is prone to coincidence errors at concentration as low as 200 cm$^{-3}$ (Lance et al., 2010), in this case an instrumental artifact cannot be ruled out without further investigation.*

Fig 11 caption. at the cloud base level? Done.

l559. Unambiguous? It is actually ambiguous. No change.

l563. I think it's better to start a new sentence for the text in parenthesis. Similar for l588 and elsewhere. Parentheses have been removed.

Figure 17 caption. It is obvious, but it might help some readers to explain the points and lines in the caption and say how the average was calculated. Done.

Figure 18 caption. (in orange) should be blue. Done.

Figure 19 caption. It would be good to describe the different lines. Done.

l857. Add "more than" before 500 /cm3? It depends on size. Done.

l894. Add (Figures B1 and B2)? Similarly for App C. Done.

**Point-by-point response to Reviewer 2's comments**

This paper describes a very valuable data set on cloud and other atmospheric properties acquired during the deployment of the French Safire ATR42 aircraft in the framework of the EUREC4A campaign out of Barbados in early 2020. In the first part the flight strategy is outlined in detail while the second part gives an overview of the measurements and the available data sets. The paper is very well written and the descriptions of the flight strategy and the data set are mostly concise yet sufficiently detailed. For the more extensive parts of the data published elsewhere respective references to more complete discussion are given. The first part laying out the flight strategy and giving the meteorological context of each flight is very helpful for the interpretation and further use of the data. The data is well structured with extensive meta data and accessible in a database. I recommend the paper and data to be published after addressing a few very minor points.

Throughout the paper acronyms should be spelled out at the first use more consistently. In addition, a table of acronyms might be useful to guide the reader. Instrument names could also be listed together with a reference to a published characterization if available. We tried to spell out acronyms more consistently throughout the paper, and a Table of acronyms has been added in Appendix. The references associated with the different instruments are given in the text.

In some sections the typesetting of mathematical symbols in italics needs to be cleaned up. Done.

P20/l303f: From the wording of the sentence it is not entirely clear if the calibration parameters for the Licor humidity data were assumed to be constant for all flights or if this was also verified. We now make the sentence less ambiguous by writing: *The Licor humidity measurements (in $g\,m^{-3}$) are calibrated against the WVSS-II absolute humidity measurements of RF13, and the same calibration coefficients are used in all flights (note however that the calibration of humidity in the SAFIRE-TURB dataset is performed leg by leg as described by Brilouet et al. (2021)).*

P22/l351: The sentence seems incomplete. "..often appear dark because the choice was made..." Do the authors mean the choice of exposure time? Done.

P25/l386: Platform should be spelled in English (maybe give the French name in the parenthesis). Done.

P25: Where the time series of the in-situ data synchronized to the ATR GPS time accounting for potential plumbing delays due to different lengths sample lines? No, not for the aerosol and cloud instruments, at least. All instruments but the UHSAS are open path instruments with fast electronics (i.e. negligible response time). The UHSAS might be prone to plumbing delays because ambient air is sucked into the instrument through a 15 cm long (estimates) 1/8" OD tube. Operated at a constant flowrate of 50 sccm, we roughly estimate the plumbing delay to be in the order of a second. It is not taken into account for now. We added to the text: *All instruments but the UHSAS are open*

*path instruments with fast electronics and therefore their response time is negligible. The potential plumbing delay of the UHSAS (estimated to be of the order of a second) is not taken into account for now.*

Sec 3.4: How was the size calibration of the aerosol and could instruments performed? Was the stability of the calibration checked in the field after each flight? The calibration of the CDP-2 was already discussed in the paper. We completed our description of the calibration process of other instruments as *The UHSAS-A used in EUREC$^4$A was last maintained and calibrated by DMT in December 2018 (using National Institute of Standards and Technology traceable polystyrene spheres of nominal diameter 100 nm) and a calibration check (using polystyrene beads of various sizes, e.g. ThermFisher Scientific 3150A) was performed at SAFIRE prior to the campaign in May 2019.* and *The calibration of the 2D-S probe was tested before the campaign with opaque calibrated features printed on glass spinning disks..*

P29/Fig 10.: The authors should consider reporting also (or exclusively?) the median to describe the statistics of the various quantities. Using the mean but an 10-90 percentiles seems inconsistent and given that the distributions are not Gauss-distributed the median is more meaningful. Actually, the caption was wrong; the quantity reported was already the median (and the 10-90 percentiles in brackets). It has been corrected.

P44/l784: I am not sure "diagnosed" is the right word here. Maybe "derived" or "deduced"? l786: see above. Done.

P45/Fig 18, middle panel: Should the lines be drawn in different colors? It might be good to include the cloud-only case also into the legend of the middle panel for clarity. We prefer using one color for the cloud-only estimates, and another color for the cloud + drizzle estimates so that it is easier to assess the ressemblances and differences among the cloud-only and cloud+drizzle quantities. On the other hand, we repeated the cloud-only legend in the middle panel as suggested.

P48/Fig 19: It is not clear what the different colors indicate. It would be good to add a legend to this plot or use a consistent color scheme for all panels. Done.